# Diversified Recommendations for Agents with Adaptive Preferences

**Arpit Agarwal**[*]
Department of Computer Science
Columbia University
New York, NY 10027
arpit.agarwal@columbia.edu

**William Brown**[†]
Department of Computer Science
Columbia University
New York, NY 10027
w.brown@columbia.edu

## Abstract

When an Agent visits a platform recommending a menu of content to select from, their choice of item depends not only on immutable preferences, but also on their prior engagements with the platform. The Recommender's primary objective is typically to encourage content consumption which optimizes some reward, such as ad revenue, but they often additionally aim to ensure that a sufficiently wide variety of content is consumed by the Agent over time. We formalize this problem as an adversarial bandit task. At each step, the Recommender presents a menu of $k$ (out of $n$) items to the Agent, who selects one item in the menu according to their unknown *preference model*, which maps their history of past items to relative selection probabilities. The Recommender then observes the Agent's selected item and receives bandit feedback of the item's (adversarial) reward. In addition to optimizing reward from the selected items at each step, the Recommender must also ensure that the total distribution of chosen items has sufficiently high entropy.

We define a class of preference models which are *locally learnable*, i.e. behavior over the entire domain can be estimated by only observing behavior in a small region; this includes models representable by bounded-degree polynomials as well as functions with a sparse Fourier basis. For this class, we give an algorithm for the Recommender which obtains $\tilde{O}(T^{3/4})$ regret against all item distributions satisfying two conditions: they are sufficiently diversified, and they are *instantaneously realizable* at any history by some distribution over menus. We show that these conditions are closely connected: all sufficiently high-entropy distributions are instantaneously realizable at any history of selected items. We also give a set of negative results justifying our assumptions, in the form of a runtime lower bound for non-local learning and linear regret lower bounds for alternate benchmarks.

## 1 Introduction

Suppose you manage an online platform that repeatedly provides menus of recommended content to visitors, such as sets of videos to watch or items to purchase, aiming to display options which agents will engage favorably with and yield you high rewards (in the form of ad revenue, watch time, purchases, or other metrics). In many settings, the preferences of agents are not fixed *a priori*, but rather can *change* as a function of their consumption patterns—the deeper one goes down a content "rabbit hole", the further one might be likely to keep going. This "rabbit hole" effect can lead to (unforeseen) loss of revenue for the platform, as advertisers may later decide that they are not willing to pay as much for this "rabbit hole" content as they would for other content. The scope of negative

---

[*]http://www.columbia.edu/ aa4931/

[†]wibrown.github.io

36th Conference on Neural Information Processing Systems (NeurIPS 2022).

effects emerging from these feedback loops is large, ranging from the emergence of "echo chambers" [15] and rapid political polarization [23] to increased homogeneity which can decrease agent utility [9], amplify bias [20], or drive content providers to leave the platform [22]. These are harms which many platforms aim to avoid, both for their own sake and out of broader societal concerns.

Hence, the evolving preferences of the Agent can be directly at odds with the Recommender's objectives of maximizing revenue and ensuring diverse consumption patterns in this *dynamic* environment. Our goal is to study such tensions between the interaction of these two players: the Recommender that recommends menus based on past choices of the Agent so as to maximize its reward (subject to diversity constraints), and the Agent whose preferences evolve as a function of past recommendations.

To this end, we consider a stylized setting where the Recommender is tasked with providing a *menu* of $k$ recommended items (out of $n$ total) every round to an Agent for $T$ sequential rounds. In each round, the Agent observes the menu, then selects one of the items according to their *preference model* $M$, which the Recommender does not know in advance. The preference model $M$ takes as input the Agent's *memory vector* $v$, which is the normalized histogram of their past chosen items, and assigns relative selection probabilities to each item. The selected item at each round results in a reward for the Recommender, specified by an adversarial sequence of reward vectors, which the Recommender receives as bandit feedback, in addition to observing which item was selected. The Recommender must choose a sequence of menus to maximize their reward (or minimize regret), subject to a *diversity constraint*, expressed as a minimum entropy for the empirical item distribution.

However, any regret minimization problem is incomplete without an appropriate benchmark for comparing the performance of a learner. An entropy constraint alone is insufficient to define a such benchmark. Due to intricacies of the Agent's preference model, there may be item distributions which are impossible to induce under any sequence of menus (e.g. they may strongly dislike the most profitable content). Adding to the challenge is the fact that the preference model is initially unknown and must be *learned*, and the set of item distributions which are *instantaneously realizable* by sampling a menu from some distribution can shift each round as well. Several immediate proposals are infeasible: it is impossible to obtain sublinear regret against the best fixed menu distribution, or even against the best item distribution realizable from the uniform memory vector. We propose a natural benchmark for which regret minimization becomes possible: the set of item distributions which are *everywhere* instantaneously realizable (the EIRD($M$) set), i.e. item distributions such that, at any memory vector, there is always *some* menu distribution which induces them. We show that this set is also closely related to entropy constraints: when $M$ is sufficiently *dispersed* (a condition on the minimum selection probability for each item), EIRD($M$) contains all sufficiently high-entropy distributions, and so regret minimization can occur over the entire high-entropy set.

## 1.1 Our Results

We give an algorithm which, for a minimum entropy set $H_c$ and preference model $M$, allows the Recommender to obtain $\tilde{O}(T^{3/4})$ regret against the best distribution in the intersection of $H_c$ and EIRD($M$), provided that $M$ satisfies $\lambda$-*dispersion* and belongs to a class $\mathcal{M}$ which is *locally learnable*. A $\lambda$-dispersed preference model $M$ assigns a preference score of at least $\lambda > 0$ to every item, ensuring a minimum positive probability of selection to each item in a menu. Dispersion is a natural assumption, given our restriction to EIRD($M$), as items which only have positive selection probability in part of the domain cannot be induced everywhere. The local learnability condition for a model class enforces that the behavior of any particular model can be predicted by observing behavior only in a small region. This is essentially necessary to have any hope of model estimation in this setting: we show that if learning a class from *exact* queries requires making queries to many points which are pairwise well-separated, exponentially many rounds are required to implement query learning. Despite this restriction, we show that several rich classes of preference models are indeed locally learnable, including those where preference scoring functions are expressed by bounded-degree multivariate polynomials, or by univariate functions with a sparse Fourier basis.

Our algorithm is explicitly separated into learning and optimization stages. The sole objective for the learning stage is to solve the *outer problem*: recover an accurate hypothesis for the preference model. We select sequences of menus which move the Agent's memory vector to various points near the uniform distribution, enabling us to implement local learning and produce a model hypothesis $\hat{M}$. We then shift our focus to the *inner problem* for the Recommender, which is natural to view as a bandit linear optimization problem over the set of distributions in consideration, as we can use $\hat{M}$ to identify

a distribution of menus which generates a particular item distribution. However, representing the $\text{EIRD}(\hat{M})$ set explicitly is impractical, as the functions which generate feasible sets from the history can be highly non-convex. Instead, we operate over the potentially larger set where intersections are taken only over the sets $\text{IRD}(v, \hat{M})$ of instantaneously realizable distributions we have observed thus far. This precludes us from using off-the-shelf bandit linear optimization algorithms as a black box, as they typically require the decision set to be specified in advance. We introduce a modification of the FKM algorithm [13], RC-FKM, which can operate over contracting decision sets, and additionally can account for the imprecision in $\hat{M}$ when generating menu distributions. This enables the Recommender to guide the Agent to minimize regret on their behalf via the sequence of menus they present.

## 1.2 Summary of Contributions

Briefly, our main contributions are:

1. We formulate the dynamic interaction between a Recommender and an Agent as an adversarial bandit task. We show that no algorithm can obtain $o(T)$ regret against the best menu distribution, or against the best item distribution in the $\text{IRD}$ set of uniform vector. We then consider $\text{EIRD}(M)$ and argue that it is a natural benchmark for regret as it also contains all sufficiently high entropy distributions over items.

2. We define a class of *locally learnable* functions, which are functions that can be learned only using samples from a small neighborhood. We show a number of rich classes of functions where this is possible, and further we show that any class which is *not* locally learnable cannot be learned quickly by any algorithm which fits a hypothesis using queries.

3. We give an algorithm for the Recommender that achieves $\widetilde{O}(T^{3/4})$ regret against $\text{EIRD}(M)$ for locally learnable classes of preference models that are $\lambda$-dispersed, which implements local learning to obtain a sufficiently accurate hypothesis for use in optimizing menu distributions. As a component of this, we develop a new algorithm for bandit linear optimization which can operate over contracting decision sets, and which can account for bounded adversarial imprecision in the played action.

Overall, by considering this stylized setting we are able to provide several insights into the dynamic interaction between an Agent and a Recommender. While our algorithm is a useful tool for a Recommender who is already committed to providing diversified recommendations, we also view our results as presenting an intrinsic argument for incorporating such constraints. When preferences adapt over time, and Agents may be prone to venturing down content "rabbit holes", restricting attention to recommendation patterns which are not too concentrated on small sets of items can in fact make the regret minimization problem tractable by discouraging consumption patterns which may be difficult to draw the Agent back from. This suggests a synergy between the goal of regret minimization and showing diverse content to the user.

## 1.3 Related Work

Feedback loops in user preferences have received significant attention in the recommender systems literature, particularly for models with multiple agents which make use of collaborative filtering methods, and with explicit adaptivity models which are less flexible than those we consider [7, 9, 20, 28, 22]. Within the online learning literature, our formalization bears some resemblance to bandit problems where multiple arms can be pulled simultaneously, which have received much recent attention [30, 29, 8, 3]. Our results also share similarities with work on optimization from revealed preferences, where a mapping to a nested convex problem must be learned [27, 12]; with the performative prediction literature, where actions induce a distribution shift which impacts instantaneous reward potential [24, 18]; and more broadly, with repeated game problems against adaptive agents [5, 11, 10]. Further related work is discussed in Appendix A.

## 1.4 Organization

In Section 2, we introduce our setting and key definitions, analyze the local learnability of several classes of preference models, and give a series of negative and structural results. In Section 3 we

introduce a bandit linear optimization algorithm for contracting sets, which we use as a subroutine for our main algorithm in Section 4. We discuss the intuition for our proof techniques throughout, with full proofs deferred to the appendix.

## 2 Model and Preliminaries

The central object of our setting is the *preference model* of the Agent, which dictates their relative item preferences based on their selection history and expresses their adaptivity over time.

**Definition 1** (Preference Models). *A preference model is a mapping $M : \Delta(n) \to [0,1]^n$ which maps memory vectors $v$ to a preference score vector $s_v = M(v)$.*

We assume that any input $v \notin \Delta(n)$ to $M$ (such as the empty history at $t = 1$) results in the uniform score vector where $M(v)_i = 1$ for all $i$. A constraint on our sequence of interactions with the Agent is that the resulting item distribution must have sufficiently high entropy.

**Definition 2** (Diversity Constraints). *A diversity constraint $H_c \subset \Delta(n)$ is the convex set containing all item distributions $v \in \Delta(n)$ with entropy at least c, i.e. $v$ is in $H_c$ if and only if:*

$$H(v) = -\sum_{i=1}^{n} v_i \log(v_i) \geq c.$$

We say that a constraint $H_c$ is $\epsilon$-*satisfied* by a distribution $v$ if we have that $\min_{x \in H_c} d_{TV}(x, v) \leq \epsilon$, where $d_{TV}$ is the total variation distance between probability distributions.

Our algorithmic results can be extended to any convex constraint set which contains a small region around the uniform distribution, but we focus on entropy constraints as they are quite natural and have interesting connections to our setting which we consider in Section 2.3.

### 2.1 Recommendation Menus for Adaptive Agents

An instance of our problem consists of an item set $N = [n]$, a menu size $k$, a preference model $M$ for the Agent, a constraint $H_c$, a horizon length of $T$ rounds, and a sequence of linear reward functions $\rho_1, \ldots, \rho_T$ for the Recommender. In each round $t \in \{1, \ldots, T\}$:

- The Recommender chooses a menu $K_t \subset N$ with $|K_t| = k$.
- The Agent chooses item $i \in K_t$ with probability

$$p_{K_t, v_t, i_t} = \frac{s_{v_t, i_t}}{\sum_{j \in K_t} s_{v_t, j}}$$

  and updates its memory vector to the normalized histogram

$$v_{t+1} = \frac{e_i}{t+1} + \frac{t \cdot v_t}{t+1},$$

  where $e_i$ is the $i$th standard unit vector.
- The Recommender observes receives reward $\rho_t(e_i)$ for the chosen item.

The goal of the Recommender is to maximize their reward over $T$ rounds subject to $v_T$ satisfying $H_c$. It might seem to the reader that the Recommender can 'manipulate' the Agent to achieve any preference score vector over time; however, this is not true as many score vectors might not be achievable depending on the preference model.

### 2.2 Realizability Conditions for Item Distributions

For any memory vector $v$, we define the feasible set of item choice distributions for Agent in the current round, each generated by a distribution over menus which the Recommender samples from.

**Definition 3** (Instantaneously-Realizable Distributions at $v$). *Let $p_{K,v} \in \Delta(n)$ be the item distribution selected by an Agent presented with menu $K$ at memory vector $v$, given by:*

$$p_{K,v,i} = \frac{s_{v,i}}{\sum_{j \in K} s_{v,j}}.$$

*The set of instantaneously-realizable distributions at $v$ is given by:*

$$\text{IRD}(v, M) = \underset{K \in \binom{n}{k}}{\text{convhull}} \, p_{K,v}.$$

For any $x \in \text{IRD}(v, M)$, any menu distribution $z \in \Delta\left(\binom{n}{k}\right)$ specifying a convex combination of menu score vectors $p_{K,v}$ which sum to $x$ will generate the item distribution $x$ upon sampling.

One might hope to match the performance of the best menu distribution, or perhaps the best realizable item distribution from the uniform vector. Unfortunately, neither of these are possible.

**Theorem 1.** *There is no algorithm which can obtain $o(T)$ regret against the best item distribution in the* IRD *set for the uniform vector, or against the best menu distribution in $\Delta\left(\binom{n}{k}\right)$, even when the preference model is known exactly and is expressible by univariate linear functions.*

We give a separate construction for each claim, with the full proof deferred to Appendix . The first is a case where the optimal distribution from the uniform vector cannot be played every round, as it draws the the memory vector into IRD sets where the reward opportunities are suboptimal. The second considers menu distributions where obtaining their late-round performance requires committing early to an irreversible course of action. Instead, our benchmark will be the set of distributions which are realizable from *any* memory vector.

**Definition 4** (Everywhere Instantaneously-Realizable Distributions). *For a preference model $M$, the set of everywhere instantaneously-realizable distributions is given by:*

$$\text{EIRD}(M) = \bigcap_{v \in \Delta(n)} \text{IRD}(v, M).$$

*This is the set of distributions $x \in \Delta(n)$ such that from any memory vector $v$, there is some menu distribution $z$ such that sampling menus from $d$ induces a choice distribution of $x$ for the agent.*

Note that the set $\text{EIRD}(M)$ is convex, as each $\text{IRD}(v, M)$ is convex by construction.

## 2.3 Conditions for Preference Models

The algorithm we present in Section 4 requires two key conditions for a class of preference models: each model in the class must be *dispersed*, and the class must be *locally learnable*. This enforces that the Agent is always willing to select every item in the menu they see with some positive probability, and that the behavior at any memory vector can be estimated by observing behavior in a small region.

**Definition 5** (Dispersion). *A preference model $M$ is $\lambda$-dispersed if $s_{v,i} \geq \lambda$ for all $v \in \Delta(n)$ and for all $i$, i.e. items always have a score of at least $\lambda$ at any memory vector.*

The dispersion condition plays an important role in the analysis of our algorithm by enabling efficient exploration, but it additionally coincides with diversity constraints in appropriate regimes.

**Theorem 2** (High-Entropy Containment in EIRD). *Consider the diversity constraint $H_c$ for $c = \log(n) - \gamma$, and let $\tau \geq \exp(-\gamma)$. Let $M$ be a $\lambda$-dispersed preference model with $\lambda \geq \frac{k^2 \exp(\gamma/\tau)}{n}$. For any vector $v \in H_c$, there is a vector $v' \in \text{EIRD}(M)$ such that $d_{TV}(v, v')$ is at most $O(\tau)$.*

The key step here, proved in Appendix B.2, is that $\text{EIRD}(M)$ contains the uniform distribution over any large subset of items, and taking mixtures of these can approximate any high-entropy distribution.

Next, for a class of models to be locally learnable, one must be able to accurately estimate a model's preference scores everywhere when only given access to samples in an arbitrarily small region.

**Definition 6** (Local Learnability). *Let $\mathcal{M}$ be a class of preference models, and let*

$$\text{EIRD}(\mathcal{M}) = \bigcap_{M \in \mathcal{M}} \text{EIRD}(M).$$

*Let $v^*$ be a point in $\text{EIRD}(\mathcal{M})$, and $V_\alpha$ be the set of points within distance $\alpha$ from $v^*$, for $\alpha$ such that $V_\alpha \subseteq \text{EIRD}(\mathcal{M})$. $\mathcal{M}$ is $h$-locally learnable if there is some $v^*$ and an algorithm $\mathcal{A}$ which, for any $M \in \mathcal{M}$ and any $\alpha > 0$, given query access to normalized score estimates $\hat{s}_v$ where $\|\hat{s}_v - M(v)/M_v^*\|_\infty \leq \beta$ for any $v \in V_\alpha$ (where $M_v^* = \sum_i M(v)_i$) and for some $\beta$, can produce a hypothesis model $\hat{M}$ such that $\left\|\hat{M}(x)/\hat{M}_x^* - M(x)/M_x^*\right\| \leq \epsilon$ for any $x \in \Delta(n)$ and $\epsilon = \Omega(\beta)$.*

The local learnability condition, while covering many natural examples shown in Section 2.4, is indeed somewhat restrictive. In particular, it is not difficult to see that classes of piecewise functions, such as neural networks with ReLU activations, are not locally learnable. However, this appears to be essentially a necessary assumption for efficient learning, given the cumulative nature of memory in our setting. We show a runtime lower bound for any algorithm that hopes to learn an estimate $\hat{M}$ for the preference model $M$ via *queries*. Even a Recommender who can force the Agent to pick a particular item each round, and exactly query the preference model for free at the current memory vector, may require exponentially many rounds to learn $\hat{M}$ if the points it must query are far apart.

**Theorem 3** (Query Learning Lower Bound). *Suppose the Recommender can force the Agent to select any item at each step $t$, and can query $M(v_t)$ at the current memory vector $v_t$. Let $\mathcal{A}_S$ be an algorithm which produces a hypothesis $\hat{M}$ by receiving queries $M(v)$ for each $v \in S$. For points $v$ and $v'$, let $d_{\max}(v, v') = \max_i v_i - v'_i$. Then, any sequence of item selections and queries by the Recommender requires at least*

$$T \geq \min_{\sigma \in \pi(S)} \prod_{i=1}^{|S|-1} (1 + d_{\max}(\sigma(i), \sigma(i+1)))$$

*rounds to run $\mathcal{A}(S)$, where $\pi(S)$ is the set of permutations over $S$ and $\sigma(i)$ is the $i$th item in $\sigma$.*

We prove this in Appendix B.3. Notably, this implies that if $S$ contains $m$ points which, for any pair $(v, v')$ have both $d_{\max}(v, v') \geq \gamma$ and $d_{\max}(v', v) \geq \gamma$, at least $O\left((1 + \gamma)^m\right)$ rounds are required.

## 2.4 Locally Learnable Preference Models

There are several interesting examples of model classes which are indeed locally learnable, which we prove in Appendix C. In general, our approach is to query a grid of points inside the radius $\alpha$ ball around the uniform vector, estimate each function's parameters and show that the propagation of over the entire domain is bounded. Note that the normalizing constants for each query we observe may differ; for univariate functions, we can handle this by only moving a subset of values at a time, allowing for renormalization. For multivariate polynomials, we consider two distinct classes and give a separate learning algorithm for each; we can estimate ratios of scores directly for multilinear functions, and if scores are already normalized we can avoid rational functions altogether. Each local learning result we prove involves an algorithm which makes queries near the uniform vector. We later show in Lemma 4 that taking $\lambda \geq k^2/n$ suffices to ensure that these queries can indeed be implemented via an appropriate sequence of menu distributions for any $M$ in such a class.

### 2.4.1 Bounded-Degree Univariate Polynomials

Let $\mathcal{M}_{BUP}$ be the class of *bounded-degree univariate polynomial* preference models where:

- For each $i$, $M(v)_i = f_i(v_i)$, where $f_i$ is a degree-$d$ univariate polynomial which takes values in $[\lambda, 1]$ over the range $[0, 1]$ for some constant $\lambda > 0$.

Univariateness captures cases where relative preferences for an item depend only on the weight of that item in the agent's memory, i.e. there are no substitute or complement effects between items.

**Lemma 1.** $\mathcal{M}_{BUP}$ is $O(d)$-locally learnable by an algorithm $\mathcal{A}_{BUP}$ with $\beta \leq O(\epsilon \lambda^2 \cdot (\frac{\alpha}{nd})^d)$.

### 2.4.2 Bounded-Degree Multivariate Polynomials

Let $\mathcal{M}_{BMLP}$ be the class of *bounded-degree multilinear polynomial* preference models where:

- For each $i$, $M(v)_i = f_i(v)$, where $f_i$ is a degree-$d$ multilinear (i.e. linear in each item) polynomial which takes values in $[\lambda, 1]$ over $\Delta(n)$ for some constant $\lambda > 0$,

and let $\mathcal{M}_{BNMP}$ be the class of *bounded-degree normalized multivariate polynomial* preference models where:

- For each $i$, $M(v)_i = f_i(v)$, where $f_i$ is a degree-$d$ polynomial which takes values in $[\lambda, 1]$ over $\Delta(n)$ for some constant $\lambda > 0$, where $\sum_i f_i(v) = C$ for some constant $C$.

Together, these express a large variety of adaptivity patterns for preferences which depend on frequencies of many items items simultaneously. In particular, these can capture relatively intricate "rabbit hole" effects, in which some subsets of items are mutually self-reinforcing, and where their selection can discourage future selection of other subsets.

**Lemma 2.** $\mathcal{M}_{BMLP}$ and $\mathcal{M}_{BNMP}$ are both $O(n^d)$-locally learnable, for $\beta \leq O(\frac{\epsilon^2}{\text{poly}(n(d/\alpha)^d))})$, and $\beta \leq \frac{\epsilon}{\alpha^d F(n,d)}$, respectively, where $F(n,d)$ is independent of other parameters.

### 2.4.3 Univariate Functions with Sparse Fourier Representations

We can also allow for classes of functions where the minimum allowable $\alpha$ depends on some parameter. Functions with sparse Fourier representations are such an example, and naturally capture settings where preferences are somewhat cyclical, such as when an Agent goes through "phases" of preferring some type of content for a limited window. We say that a function $f : \mathbb{R} \to \mathbb{R}$ is $\ell$-sparse if $f(x) = \sum_{i=1}^{\ell} \xi_i e^{2\pi \mathbf{i} \eta_i x}$ where $\eta_i \in [-F, F]$ denotes the $i$-th frequency and $\xi_i$ denotes the corresponding magnitude. We say that an $\ell$-sparse function $f$ is $\hat{\alpha}$-separable when $\min_{i \neq j} |\eta_i - \eta_j| > \hat{\alpha}$. Let $\mathcal{M}_{SFR}(\hat{\alpha})$ be the class of univariate *sparse Fourier representation* preference models where:

- For each $i$ , $M(v)_i = f_i(v_i)$, where $f_i$ is a univariate $\ell$-sparse and $\hat{\alpha}$-separable function which, over $[0, 1]$, is $L$-Lipschitz and takes values in $[\lambda, 1]$ for some constant $\lambda > 0$.

**Lemma 3.** $\mathcal{M}_{SFR}(\hat{\alpha})$ is $\widetilde{O}(n\ell)$-locally learnable by an algorithm $\mathcal{A}_{SFR}$ with $\beta \leq O(\frac{\epsilon\lambda\alpha}{\sqrt{n\ell}})$ and any $\alpha \geq \tilde{\Omega}(1/\hat{\alpha})$.

## 3 Bandit Linear Optimization with Contracting Sets

The inner problem for the Recommender can be viewed as a bandit linear optimization problem over $H_c \cap \text{EIRD}(M)$. However, representing $\text{EIRD}(M)$ is challenging even if we know $M$ exactly, as it involves an intersection over infinitely many sets (generated by a possibly non-convex function), and a net approximation would involve exponential dependence on $n$. Instead, our approach will be to operate over the larger set $H_c \cap (\bigcap_t \text{IRD}(v_t, M))$ for the memory vectors $v_t$ we have seen thus far, where representing each $\text{IRD}$ has exponential dependence only on $k$ (from enumerating all menus).

The tradeoff is that we can no longer directly use off-the-shelf bandit linear optimization algorithms for a known and fixed decision set such as FKM [13] or SCRIBLE [1] as a subroutine, as our decision set is contracting each round. We introduce an algorithm for bandit linear optimization, a modification of the FKM algorithm we call Robust Contracting FKM (RC-FKM), which handles this issue by projecting to our estimate of the contracted decision set at each step. Additionally, RC-FKM can handle the imprecision resulting from our model estimation step, which can be represented by small adversarial perturbations to the action vector in each round; we modify the sampling rule to ensure that our target action remains in the true decision set even when perturbations are present. We prove the regret bound for RC-FKM in Appendix D.

---

**Algorithm 1** (Robust Contracting FKM).

---

Input: sequence of contracting convex decision sets $\mathcal{K}_1, \ldots \mathcal{K}_T$ containing $\mathbf{0}$, perturbation vectors $\xi_1, \ldots, \xi_T$ where $\|\xi_t\| \leq \epsilon$, parameters $\delta, \eta$.
Set $x_1 = \mathbf{0}$
**for** $t = 1$ to $T$ **do**
    Draw $u_t \in \mathbb{S}_1$ uniformly at random, set $y_t = x_t + \delta u_t + \xi_t$
    Play $y_t$, observe and incur loss $\phi_t \in [0, 1]$, where $\mathbb{E}[\phi_t] = f_t(y_t)$
    Let $g_t = \frac{n}{\delta}\phi_t u_t$
    Let $\mathcal{K}_{t+1,\delta,\epsilon} = \{x | \frac{r}{r-\delta-\epsilon} x_t \in \mathcal{K}_{t+1}\}$
    Update $x_{t+1} = \Pi_{\mathcal{K}_{t+1,\delta,\epsilon}}[x_t - \eta g_t]$
**end for**

---

**Theorem 4** (Regret Bound for Algorithm 1). *For a sequence of $G$-Lipschitz linear losses $f_1, \ldots, f_T$ and a contracting sequence of domains $\mathcal{K}_1, \ldots, \mathcal{K}_T$ (with $\mathcal{K}_j \subseteq \mathcal{K}_i$ for $j > i$, each with diameter at most $D$, and where a ball of radius $r > \delta + \epsilon$ around $\mathbf{0}$ is contained in $\mathcal{K}_T$), and adversarially*

*chosen unobserved vectors $\xi_1, \ldots, \xi_T$ with $\|\xi_t\| \leq \epsilon$ which perturb the chosen action at each step, with parameters $\eta = \frac{D}{nT^{3/4}}$ and $\delta = \frac{1}{T^{1/4}}$, Algorithm 1 obtains the expected regret bound*

$$\sum_{t=1}^{T} \mathbb{E}[\phi_t] - \min_{x \in \mathcal{K}_T} \sum_{t=1}^{T} f_t(x) \leq nGDT^{3/4} + \frac{GDT^{3/4}}{r} + \frac{2\epsilon GDT}{r}.$$

## 4  Recommendations for Adaptive Agents

Our main algorithm begins with an explicit learning phase, after which we conduct regret minimization, and at a high level works as follows:

- First, we learn an estimate of the preference model $\hat{M}$ by implementing local learning with a set of points close to the uniform memory vector, which suffices to ensure high accuracy of our representation with respect to $M$. If the number of local learning queries is independent of error terms and $\beta = \Theta(\epsilon)$, we can complete this stage in $t_0 = \tilde{O}(1/\epsilon^3) = \tilde{O}(T^{3/4})$ steps.

- For the remaining $T - t_0$ steps, we implement RC-FKM by using the learned model $\hat{M}$ at each step to solve for a menu distribution which generates the desired item distribution from the current memory vector, then contracting the decision set based on the memory update.

**Theorem 5** (Regret Bound for Algorithm 2). *Algorithm 2 obtains regret bounded by*

$$\text{Regret}_{C \cap \texttt{EIRD}(M)}(T) \leq \tilde{O}\left(t_0 + nGT^{3/4} + \frac{(\delta + \epsilon)GT}{r} + \epsilon GT\right) = \tilde{O}(T^{3/4})$$

*where $t_0$ is the time required for local learning, $r = O(k^2/n)$, and $\epsilon, \delta = O(r \cdot T^{-1/4})$, and results in an empirical distribution such that $H_c$ is $O(\epsilon)$-satisfied with probability at least $1 - O(T^{-1/4})$.*

### 4.1  Structure of `EIRD`$(M)$

The key tool which enables us to implement local learning is a construction for generating any point near the uniform via an adaptive sequence of menu distributions, provided $\lambda$ is sufficiently large.

**Lemma 4.** *For any $\lambda$-dispersed $M$ where $\lambda \geq \frac{k^2}{n}$, `EIRD`$(M)$ contains all points $x \in \Delta(N)$ satisfying*

$$\|x - x_U\|_\infty \leq \frac{k - 1}{n(n - 1)},$$

*where $x_U$ is the uniform $\frac{1}{n}$ vector.*

We give an algorithmic variant of this lemma which is used directly by Algorithm 2, as well as a variant for uniform distributions over smaller subsets as $\lambda$ grows, which we use to prove Theorem 2.

### 4.2  Subroutines

Our algorithm makes use of a number of subroutines for navigating the memory space, model learning, and implementing RC-FKM. We state their key ideas here, with full details deferred to Appendix E.

`UniformPad`:

- In each round, include the $k$ items with smallest counts, breaking ties randomly.

`MoveTo`$(x)$:

- Apply the same approach from `UniformPad` to the difference between the current histogram and $x$.

`Query`$(x)$:

- Play a sequence of $O(n/k)$ partially overlapping menus which cover all items, holding each constant long enough for concentration, and compute relative probabilities of each item.

**Algorithm 2** A no-regret recommendation algorithm for adaptive agents.

---

Input: Item set $[n]$, menu size $k$, Agent with $\lambda$-dispersed memory model $M$ for $\lambda \geq \frac{k^2}{n}$, where $M$ belongs to an $S$-locally learnable class $\mathcal{M}$, diversity constraint $H_c$, horizon $T$, $G$-Lipschitz linear losses $\rho_i, \ldots, \rho_T$.

Let $t_{\text{pad}} = \tilde{\Theta}(1/\epsilon^3)$

Let $t_{\text{move}} = \tilde{\Theta}(1/\epsilon^3)$

Let $t_{\text{query}} = \tilde{\Theta}(1/\epsilon^2)$

Let $\alpha = \Theta(\frac{k}{n^2 S})$

Get set of $S$ points in the $\alpha$-ball around uniform vector $x_U$ to query from $\mathcal{A}_{\mathcal{M}}$

Let $t_0 = t_{\text{pad}} + S(2 \cdot t_{\text{move}} + t_{\text{query}})$

Run `UniformPad` for $t_{\text{pad}}$ rounds

**for** $x_i$ in $S$ **do**

    Run `MoveTo`$(x_i)$ for $t_{\text{move}}$ rounds

    Run `Query`$(x_i)$ for $t_{\text{query}}$ rounds, observe result $\hat{q}(x_i)$

    Run `MoveTo`$(x_U)$ for $t_{\text{move}}$ rounds

**end for**

Estimate model $\hat{M}$ using $\mathcal{A}_{\mathcal{M}}$ for $\beta = \Theta(\epsilon)$

Let $v_{t_0}$ be the empirical item distribution of the first items $t_0$ items

Let $\mathcal{K}_{t_0} = H_c$ (in $n-1$ dimensions, with $x_{t,n} = 1 - \sum_{i=1}^{n-1} x_{t,i}$, and s.t. $x_U$ translates to $\mathbf{0}$)

Initialize RC-FKM to run for $T^* = T - t_0 - 1$ rounds with $r = O(k^2/n), \delta, \epsilon = \frac{r}{T^{*1/4}}$

**for** $t = t_0 + 1$ to $T$ **do**

    Let $x_t$ be the point chosen by RC-FKM

    Use `PlayDist`$(x_t)$ to compute menu distribution $z_t$

    Sample $K_t \sim z_t$, show $K_t$ to Agent

    Observe Agent's chosen item $i_t$ and reward $\rho_t(e_{i_t})$

    Update RC-FKM with $\rho_t(e_{i_t})$

    Let $v_t = \frac{t-1}{t} v_{t-1} + \frac{1}{t} \cdot e_{i_t}$

    Update the decision set to $\mathcal{K}_{t+1} = \mathcal{K}_t \cap \text{IRD}(v_t, \hat{M})$

**end for**

---

`PlayDist`$(x)$**:**

- Given an item distribution $x$, we solve a linear program to compute a menu distribution $z_x$ using $\hat{M}(v)$ which induces $x$ when a menu is sampled and the Agent selects an item.

The intuition behind our learning stage is that each call to `Query`$(x)$ can be accurately estimated by bounding the "drift" in the memory vector while sampling occurs, as the number of samples per query is small compared to the history thus far. Each call to `MoveTo`$(x)$ for a point within the $\alpha$-ball can be implemented by generating an empirical distribution corresponding to a point in `EIRD`$(M)$ for sufficiently many rounds.

The resulting model estimate $\hat{M}$ yields score estimates which are accurate for any memory vector. To run RC-FKM, we translate to an $n-1$ dimensional simplex representation, and construct a menu distribution to implement any action $x_t$ via a linear program (`PlayDist`$(x)$). The robustness guarantee for RC-FKM ensures that the loss resulting from imprecision in $\hat{M}$ is bounded, and further ensures that the resulting expected distribution remains inside $H_c$ (and that $H_c$ is approximately satisfied with high probability by the empirical distribution). We contract our decision set in each step with the current space `IRD`$(v_t, \hat{M})$, which will always contain `EIRD`$(\hat{M})$, the best point in which is competitive with the best point in `EIRD`$(M)$.

## 5 Conclusion and Future Work

Our work formalizes a bandit setting for investigating online recommendation problems where agents' preferences can adapt over time and provides a number of key initial results which highlight the importance of diversity in recommendations, including lower bounds for more "ambitious" regret benchmarks, and a no-regret algorithm for the `EIRD` set benchmark, which can coincide with the

high-entropy set under appropriate conditions. Our results showcase a tradeoff between the space of strategies one considers and the ability to minimize regret. Crucially, our lower bound constructions illustrate that we cannot hope to optimize over the set of recommendation patterns which may send agents down "rabbit holes" that drastically alter their preferences, whereas it is indeed feasible to optimize over the space of sufficiently diversified recommendations.

There are several interesting directions which remain open for future investigation, including additional characterizations of the EIRD set, discovering more examples or applications for local learnability, identifying the optimal rate of regret or dependence on other parameters, settings involving multiple agents with correlated preferences, and consideration of alternate models of agent behavior which circumvent the difficulties posed by uniform memory.

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
