# A    Further Related Work

## A.1    Empirical Investigation Of Recommendation Feedback Loops

A substantial body of evidence has emerged in recent years indicating that recommendation systems can create feedback loops which drive negative social consequences. [23] observed that users accessing videos with extreme political views are likely to get caught in an "ideological bubble" in just a few clicks, and [16] explore the role of recommendation algorithms in creating distrust and amplifying political polarization on social media platforms. By investigating a real-world e-commerce dataset, [15] study the way in which recommendation systems drive agents' self-reinforcing preferences and lead them into "echo chambers" where they are separated from observing a diversity of content. [31] conduct a meta-analysis over many datasets which focuses specifically on the "rabbit hole" problem by means of exploring "taste distortion" of agents who observe recommendations which are more extreme than their current preferences. Such results motivate investigating these dynamics from game-theoretic and learning-theoretic foundations.

## A.2    Modeling Feedback Loops in Recommendation Systems

A number of recent works from the recommendation systems literature have explored the role of collaborative filtering algorithms for various models of agent behavior, aiming to understand how feedback loops in recommendation patterns emerge, the harms they cause, and how they can be corrected [7, 28]. A common theme is homogenization of recommendations across a *population* of users, which can lead to exacerbation of biased utility distributions for minority groups [20], long-run utility degradation [9], and a lack of traffic to smaller content providers which results in them being driven to exit the platform [22]. Our work indirectly addresses this phenomenon by encouraging diverse recommendations, but our primary focus is from the perspective of a single agent, who may be led down a "rabbit hole" by an algorithm which optimizes for their immediate engagement.

## A.3    Dueling Bandits

The "dueling bandits" problem, initially proposed as a model for similar recommendation systems challenges [30, 29], and which has been generalized for sets larger than two [3, 26], considers a similar setting in which bandit optimization is conducted with respect to the preference model of an agent, occasionally represented via an explicit parametric form. Here, one presents a set of choices to an agent, then receives only *ordinal* feedback about the relative rewards of the choices, and must optimize recommendations with regret measured against the best individual choice. In contrast to our setting, these works consider preferences which are fully determined *a priori*, and do not change as a function of item history or exhibit preference feedback loops.

## A.4    Online Stackelberg Problems

A number of works in recent years explore online problems where an agent responds to the decision-maker's actions, influencing their reward. The performative prediction setting, introduced in [24], captures settings in which a deployed classifier results in changes to the distribution itself, in turn affecting performance. This work has been extended to handle stochastic feedback [21] and notably, to a no-regret variant [18] which involves learning mapping between classifiers and distribution shifts, which bears some conceptual similarities to our procedure for locally learning an agent's preference model. The "revealed preferences" literature involves a similar requirement of learning a mapping between actions and agent choices [27, 12]. Some features of our setting resemble elements of other well-studied online problems, including the restricted exploration ability for limited switching problems (e.g. [4]), and the contracting target set for chasing nested convex bodies (e.g. [6]).

## A.5    Strategizing Against Adaptive Agents

Some recent work has begun to explore the problem of designing optimal strategies in a repeated game against agents who *adapt* their strategies over time using a no-regret algorithm. In auction problems, [5] study the extent to which an auction designer can extract value from bidders who use different kinds of no-regret algorithms. More generally, [11] connect this line of investigation to Stackelberg equilibria for normal-form games. In strategic classification problems, [32] study the

behavior when using a learning rate which is either much faster or much slower than that of the agents which one aims to classify, and draw connections to equilibrium concepts as well. Our work extends this notion of strategizing against adaptive agents to recommendation settings, with novel formulations of adaptivity and regret to suit the problem's constraints.

# B  Omitted Proofs for Sections 2

## B.1  Proof of Linear Regret Lower Bounds (Theorem 1)

We give a separate lower bound construction for the uniform IRD item distribution benchmark and the menu distribution benchmark, yielding the theorem.

**Lemma 5.** *There is no algorithm which can obtain $o(T)$ regret against the best item distribution in the* IRD *set for the uniform vector, even when the preference model is known exactly and is expressible by univariate linear functions.*

*Proof.* First we give an example for which obtaining $o(T)$ regret against $\mathtt{IRD}(v, M)$ for the uniform vector $v_U$ is impossible. Consider the memory model $M$ where:

- $M(v)_1 = \lambda + 0.5 + \frac{n}{n-1} \cdot (v_1 - \frac{1}{n}) \cdot (0.5 - \lambda)$;

- $M(v)_2 = \lambda + 0.5(1 - v_1 + \frac{1}{n})$;

- $M(v)_i = 0.5 + \lambda$ for $i > 2$.

Observe that at the uniform distribution where $v_1 = \frac{1}{n}$, all items have a score of $0.5 + \lambda$. If $v_1 = 1$, we have that:

- $M(v)_1 = 1$, and

- $M(v)_2 = \lambda + \frac{0.5}{n}$.

If $v_1 = 0$, we have that:

- $M(v)_1 = \lambda + 0.5 - \frac{0.5-\lambda}{n-1}$ , and

- $M(v)_2 = \lambda + 0.5 \cdot (1 + \frac{1}{n})$

As scores linearly interpolate between these endpoints for any $v_1$, $M$ is $\lambda$-dispersed, and scores lie in $[\lambda, 1]$. Let $k = 2$. Consider reward functions which give reward $\alpha > 0$ for item 1 in each round up to $t^* = T/2$, giving reward 0 to each other item; after $t^*$, a reward of $\beta > 0$ is given for item 2 while the rest receive a reward 0. The distribution which assigns probability $1/2$ each to item 1 and 2, with all other items having probability 0, is contained in $\mathtt{IRD}(v_U, M)$, as one can simply play the menu with both items. This distribution yields a total expected reward of

$$R_v = \frac{\alpha t^*}{2} + \frac{\beta t^*}{2}$$

over $T$ steps. Consider the performance of any algorithm $\mathcal{A}$ which results in item 1 being selected with an empirical probability $p$ over the first $t^*$ rounds. At $t = t^*$, we have $v_{t^*,1} = p$; its total reward over the first $t^*$ rounds is $\alpha p t^*$. For sufficiently large $n$ and small $\lambda$, the score for item 2 is approximated by $M(v)_2 = 0.5(1 - p)$ up to any desired accuracy. In future rounds $t \geq t^*$, the value $v_{t,s1}$ is at least $\frac{pt^*}{t}$, and so the score for item 2 is at most

$$M(v)_2 = 0.5(1 - \frac{pt^*}{t}).$$

Each other item has a score of at least 0.5, yielding an upper bound on the probability that item 2 can be selected even if it is always in the menu, as well as a maximum expected per-round reward of

$$R_t = \beta \cdot \left( \frac{0.5(1 - \frac{pt^*}{t})}{1 - 0.5\frac{pt^*}{t}} \right).$$

At time $T = 2t^*$, the instantaneous reward is at most

$$R_T = \beta \cdot \left( \frac{2-p}{4-p} \right),$$

which is also a per-round upper bound for each $t \geq t^*$. This bounds the total reward for $\mathcal{A}$ by

$$R_{\mathcal{A}} = \alpha p t^* + \beta t^* \cdot \frac{2-p}{4-p}.$$

We can now show that for any $p$, there exists a $\beta$ such that $R_v - R_{\mathcal{A}} = \Theta(T)$. For any $p \leq \frac{1}{3}$, we have

$$R_{\mathcal{A}} \leq \frac{\alpha t^*}{3} + \frac{\beta t^*}{2},$$

and for any $p > \frac{1}{3}$ we have:

$$R_{\mathcal{A}} \leq \alpha t^* + \frac{5\beta t^*}{11}.$$

In the first case, we immediately have $R_v - R_{\mathcal{A}} \geq T\alpha/6$ for any $\beta$. In the second case, let $\beta \geq 22\alpha$. We then have:

$$R_v - R_{\mathcal{A}} \geq \frac{\beta t^*}{22} - \frac{\alpha t^*}{2}$$
$$\geq T\alpha/4.$$

The value of $\beta$ can be determined adversarially, and so there is no algorithm $\mathcal{A}$ which can obtain $o(T)$ regret against $\texttt{IRD}(v, M)$.

$\square$

Next we show a similar impossibility result for regret minimization with respect to the set of all menu distributions.

**Lemma 6.** *There is no algorithm which can obtain $o(T)$ regret against the best menu distribution in $\Delta\left(\binom{n}{k}\right)$, even when the preference model is known exactly and is expressible by univariate linear functions.*

*Proof.* Let $M$ be the $\lambda$-dispersed memory model where the functions for items $(a, b, c)$, and every other item $i$, are given by:

- $M(v)_a = \lambda + (1 - \epsilon)(1 - v_b)$;

- $M(v)_b = \lambda + (1 - \epsilon)v_b$;

- $M(v)_c = \lambda + (1 - \epsilon)v_c$;

- $M(v)_i = \lambda + (1 - \epsilon)(1 - v_b)$ for $i \notin \{a, b, c\}$;

for some $\lambda > 0$ and $\epsilon > \lambda$. Let $k = 2$. Consider a sequence of rewards $\{f_t\}$ which yields reward $\alpha$ to items $(a, b)$ for each round $t \leq t^*$ and 0 to the rest, then in each step after $t^*$, yields a reward of $\beta$ for item $c$, a reward of 0 for item $b$, and reward of $-\beta$ for every other item. Note the total expected reward for the following distributions:

$$R_{(a,b)}(T) = \alpha t^* - \beta(T - t^*)/2;$$
$$R_{(b,c)}(T) = \alpha t^*/2 + \beta(T - t^*)/2;$$

The bound for $R_{(a,b)}(t^*)$ follows from symmetricity of the resulting stationary distribution, given by the unique solution $v_a = 0.5$ to the recurrence:

$$v_b = \frac{\lambda + (1 - \epsilon)v_b}{2\lambda + (1 - \epsilon)}$$

which is approached in expectation for large $T$ regardless of initial conditions for any constant $\lambda$. Symmetricity also results in balanced expectations for each item in $R_{(b,c)}$.

Consider the distribution $p_{t^*}$ played by an algorithm $\mathcal{A}$ over the first $t^*$ rounds, where $t^*$ is large enough to ensure concentration. If $p_{t^*,a} + p_{t^*,b} \leq 1 - \delta$ for some constant $\delta$, then for $\beta = 0$ the algorithm has regret $\delta \alpha t^* = \Theta(T)$ for any $t^* = \Theta(T)$. Further, if regret is not bounded, the menu $(a, b)$ must be played in nearly every round, as other item placed in the menu has positive selection probability. As such, the empirical probability of $b$ must be close to $1/2$.

After $t^*$, the algorithm cannot obtain a per-round utility which matches that of $(b, c)$ up to $\delta$ until a round $t$ where either:

$$\frac{\lambda + (1 - \epsilon)p_{t,c}}{2\lambda + (1 - \epsilon)(p_{t,b} + p_{t,c})} \geq 1/2 - \delta$$

or

$$\frac{\lambda + (1 - \epsilon)p_{t,c}}{2\lambda + (1 - \epsilon)(1 - p_{t,b} + p_{t,c})} \geq 1/2 - \delta,$$

which requires the total number of rounds in which $c$ is chosen to approach $t^*/2 - C \cdot \delta t^*$, where $C$ is a constant depending on $\epsilon$ and $\lambda$. Let $T = 3t^*/2$, and so this cannot happen for small enough constant $\delta$, resulting in a regret of $\delta \beta T/3 - \alpha T/3$ with respect to $(b, c)$, which is $\Theta(T)$ when $\delta\beta > \alpha$.

$\square$

## B.2 Proof of High-Entropy Containment of EIRD (Theorem 2)

*Proof.* By Lemma 14, for a $\lambda$-dispersed preference model $M$ with $\lambda \geq \frac{Ck^2}{n}$, any uniform distribution over $n/C$ items lies inside $\text{EIRD}(M)$. We make use of a lemma from [2], which we restate here.

**Lemma 7** (Lemma 8 in [2]). *For a random variable $A$ over $[n]$ with $H(A) \geq \log n - \gamma$, there is a set of $\ell + 1 = O(\gamma/\tau^3))$ distributions $\psi_i$ for $i \in \{0, \ldots, n\}$ over a partition of the support of $A$ which can be mixed together to generate $A$, where $\psi_0$ has weight $O(\tau)$, and where for each $i \geq 1$:*

1. *$\log |\text{supp}(\psi_i)| \geq \log n - \gamma/\tau$.*

2. *$\psi_i$ is within total variation distance $O(\tau)$ from the uniform distribution on its support.*

Using this, we can explicitly lower bound the support of each $\psi_i$:

$$\begin{aligned} \log |\text{supp}(\psi_i)| &\geq \log(n) - \gamma/\tau \\ &= \log(n) - \log(\exp(\gamma/\tau)) \\ &= \log\left(\frac{n}{\exp(\gamma/\tau)}\right). \end{aligned}$$

As such:

$$|\text{supp}(\psi_i)| \geq \frac{n}{\exp(\gamma/\tau)}.$$

Each uniform distribution over $\text{supp}(\psi_i)$ lies inside $\text{EIRD}(M)$ for $\lambda \geq \frac{Ck^2}{n}$, provided that $C \geq \exp(\gamma/\tau)$. The $O(\tau)$ bound on total variation distance is preserved under mixture, as well as when redistributing the mass of $\psi_0$ arbitrarily amongst the uniform distributions.

$\square$

## B.3 Proof of Query Learning Runtime Lower Bound (Theorem 3)

*Proof.* For any permutation $\sigma$, we can lower bound the steps required to move between any two vectors adjacent in the ordering in terms of $d_{\max}$ and the number of rounds elapsed thus far.

**Lemma 8.** *Consider two vectors $v$ and $v'$, where $v$ is the current empirical item distribution after $t$ steps. Reaching an empirical distribution of $v'$ requires at least $t \cdot d_{\max}(v, v')$ additional steps.*

*Proof.* Let $x$ be the histogram representation of $v$ with total mass $t$, and let $j^* = \arg\max_j v_j - v'_j$, where $v_j - v'_j = d_{\max}(v, v')$. Let $x' = t' \cdot v'$ be the histogram representation of $v'$ with total mass $t'$, such that $x_{j^*} = x'_{j^*}$. Note that $t'$ is the smallest total mass (or total number of rounds) where a histogram can normalize to $v'$, as any subsequent histogram must have $x'_{j^*} \geq x_{j^*}$. As such, we must have that $t' \cdot v'_{j^*} \geq t \cdot v_{j^*}$, implying that:

$$\frac{t'}{t} \geq \frac{v_{j^*}}{v'_{j^*}}$$
$$= \frac{v'_{j^*} + d_{\max}(v, v')}{v'_{j^*}}$$
$$\leq 1 + d_{\max}(v, v').$$

$\square$

At least one round is required to reach the first vector in a permutation, and we can use the above lemma to lower-bound the rounds between any adjacent vectors in the ordering. Taking the minimum over all permutations gives us the result. $\square$

## C  Proofs of Local Learnability for Section 2.4

Each proof gives a learning algorithm which operates in a ball around the uniform vector, which is contained in $\mathrm{EIRD}(M)$ whenever $\lambda \geq \frac{k^2}{n}$ by Lemma 4.

### C.1  Proof of Univariate Polynomial Local Learnability

*Proof.* Query the uniform vector $v_U$ where each $v_i = \frac{1}{n}$. Let $Z = \frac{\sqrt{nd/6}}{\alpha}$. Consider three sets each of $d/2$ memory vectors where the items with indices satisfying $i \bmod 3 = z$ each have memory values $\frac{1}{n} + \frac{j}{Z}$, items satisfying $i \bmod 3 = z + 1$ have values $\frac{1}{n} - \frac{j}{Z}$, and the remainder have $\frac{1}{n}$ (for $z \in \{0, 1, 2\}$, and for $1 \leq j \leq \frac{d}{2}$). All such vectors lie in $V_\alpha$, as $2n/3 \cdot (d/(2Z))^2 \leq \alpha^2$. Query each of the $3d/2$ vectors. For each query, let $R_v$ be the sum of all scores of the items held at $\frac{1}{n}$, divided by the sum of those same items' scores in the uniform query. Divide all scores by $R_v$. Let $R_v^*$ be the be the corresponding ratio of these sums of scores under $\{f_i\}$; each sum is within $[\frac{\lambda}{3}, 1]$ at each vector, and the sums of observed scores have additive error at most $n\beta/3$. As such, $R_v$ has additive error at most $\frac{2n\beta}{\lambda}$ from $R_v^*$. This gives us estimates for $d + 1$ points of $\hat{f}_i(x_j) = \hat{y}_j$ for each polynomial, up to some universal scaling factor. We can express this $d$-degree polynomial $\hat{f}_i$ via Lagrange interpolation:

$$L_{d,j}(x) = \prod_{k \neq j}^{d} \frac{x - x_k}{x_j - x_k};$$

$$\hat{f}_i(x) = \sum_{j=0}^{d} \hat{y}_j L_{n,j}.$$

Note that $\sum_i \hat{f}_i(v_U) = 1$ as the scores coincide exactly with our query results at the uniform vector. To analyze the representation error, let $\{f_i^*\}$ be the set of true polynomials $f_i$ rescaled to sum to 1 at the uniform vector; this involves dividing by a factor $S \in [n\lambda, n]$, and produces identical scores at every point. Consider the difference $\left|\hat{y}_j - y_j^*\right|$ for each $y_j^* = f_i^*(x_j)$. The query error for $\hat{y}_j$ prior to rescaling is at most $\beta$; rescaling by $R_v^*$ would increase this to at most $3\beta/\lambda$, which is amplified to at most

$$\left|\hat{y}_j - y_j^*\right| \leq \frac{3\beta}{\lambda} + \frac{2n\beta}{\lambda} \leq \frac{3n\beta}{\lambda}$$

as each query score is at most 1 (and our setting is trivial for $n \leq 2$). The magnitude of each of the $d+1$ Lagrange terms can be bounded by:

$$|L_{d,j}(x)| \leq \prod_{j=1}^{d/2} \frac{Z^2}{j^2}$$

$$\leq \frac{Z^d}{((d/2)!)^2}$$

for any $x \in [0,1]$, and so for any function $\hat{f}_i(x)$ we can bound its distance from $f_i^*(x)$ by:

$$\left| f_i^*(x) - \hat{f}_i(x) \right| = (d+1) \cdot \frac{3n\beta Z^d}{\lambda((d/2)!)^2}$$

$$\leq \frac{(d+1)3n\beta Z^d}{\lambda 2^{d/2}}.$$

This holds simultaneously for each $\hat{f}_i$ which, using the fact that the true ratio is at least $\lambda/n$ and the per-function bound applies to each denominator term, gives us a total bound on the score estimates we generate:

$$\left| \frac{\hat{f}_i(x)}{\sum_{j=1}^x \hat{f}_j(x)} - \frac{f_i(x)}{\sum_{j=1} f_j(x)} \right| \leq \left( 1 + \frac{(d+1)3n\beta Z^d}{\lambda 2^{d/2}} \right) \cdot \frac{(d+1)3n^3\beta Z^d}{\lambda^2 2^{d/2}}$$

$$\leq \frac{7n^3 d\beta Z^d}{\lambda^2 2^{d/2}}$$

$$\leq \frac{3 \cdot (6nd)^{d/2+2}\beta}{\alpha^d \lambda^2 2^{d/2}}$$

$$= \frac{(3nd)^{d/2+2}\beta}{\alpha^d \lambda^2}.$$

Taking $\beta \leq \frac{\epsilon \alpha^d \lambda^2}{(3nd)^{d/2+2}}$ gives us an absolute error of at most $\epsilon$ per item score, satisfying a Euclidean bound of $\epsilon$ from any true score vector $M(w)/M_w^*$ for our hypothesis $\hat{M}(v) = \{\hat{f}_i(v_i) : i \in [n]\}$. $\qquad \square$

## C.2 Proofs of Multivariate Polynomial Local Learnability

Recall that the two classes of multivariate polynomial models we consider are *bounded-degree multilinear polynomial* preference models $\mathcal{M}_{BMLP}$, where:

- for each $i$, $M(v)_i = f_i(v)$, where $f_i$ is a degree-$d$ multilinear (i.e. linear in each item) polynomial which takes values in $[\lambda, 1]$ over $\Delta(n)$ for some constant $\lambda > 0$,

and the class of *bounded-degree normalized multivariate polynomial* preference models $\mathcal{M}_{BNMP}$, where:

- for each $i$, $M(v)_i = f_i(v)$, where $f_i$ is a degree-$d$ polynomial which takes values in $[\lambda, 1]$ over $\Delta(n)$ for some constant $\lambda > 0$, where $\sum_i f_i(v) = C$ for some constant $C$.

We prove local learnability results for each case.

**Lemma 9.** $\mathcal{M}_{BMLP}$ *is* $O(n^d)$-*locally learnable by an algorithm* $\mathcal{A}_{BMLP}$ *with* $\beta \leq O(\frac{\epsilon^2}{\text{poly}(n(d/\alpha)^d)})$.

*Proof.* Consider the set of polynomials where each $v_n$ term is reparameterized as $1 - \sum_{i=1}^{n-1} v_i$, then translated so that the uniform vector appears at the origin (i.e. with $x_n = -\sum_{i=1}^{n-1} x_i$). Our approach will be to learn a representation of each polynomial normalized their sum, which is unique up to a universal scaling factor. Let $f_i^*$ be the representation of $f_i$ in this translation. Consider the $N = \sum_{j=0}^d \binom{n-1}{j}$-dimensional basis $\mathcal{B}$ where each variable in a vector $x$ corresponds to a

monomial of at most $d$ variables in $v$, each with degree 1, with the domain constrained to ensure mutual consistency between monomials, e.g.:

$$\mathcal{B} = \{1, v_1, \ldots, v_{n-1}, v_1 v_2, \ldots, \prod_{j=n-d}^{n-1} v_j\}.$$

Observe that each $f_i^*$ is a linear function in this basis. Let $q_i(x) = M(v)_i / M_v^*$ denote the normalized score for item $i$ at $v$, where $v$ translates to $x$ in the new basis. For we any $x$ we have:

$$\frac{f_i^*(x)}{\sum_{j=1}^n f_j^*(x)} = q_i(x),$$

and let $\hat{q}_i(x)$ denote the analogous perturbed query result, both of which sum to 1 over each $i$. We are done if we can estimate the vector $q(x)$ up to distance $\epsilon$ for any $x$.

With $f_i^*(x) = \langle a, x \rangle + a_0$ and $\sum_{i=1}^n f_i^*(x) = \langle b, x \rangle + b_0$, our strategy will be to estimate the ratio of each coefficient with $b_0$, for each $f_i^*$, in increasing order of degree. While our parameterization does not include item $n$, we will explicitly estimate $b$ separate from each $a$, which we can then use to estimate $f_n^*(x) = \langle b, x \rangle + b_0 - \sum_{i=1}^{n-1} f_i^*(x)$. For a monomial $m$ of degree $j$, we can estimate its coefficient for all $f_i^*$ simultaneously by moving the values for variables it contains simultaneously from the $\mathbf{0}$ vector, and viewing the restriction to its subset monomials as a univariate polynomial of degree $j$. We will use a single query to the $\mathbf{0}$ vector, and $2j + 1$ additional queries for each degree-$j$ monomial (which can be used for learning that monomial's coefficient in all $f_i^*$ simultaneously), resulting in a total query count of:

$$1 + \sum_{j=1}^d (2j+1) \cdot \binom{n-1}{j} = 1 + \sum_{j=1}^d (2j+1) \frac{(n-1)!}{j!(n-j-1)!}$$
$$= O(n^d).$$

Querying $\mathbf{0}$ gives us an estimate for each additive term:

$$\frac{\hat{a}_0^i}{b_0} = \hat{q}_i(\mathbf{0})$$

which sum to 1 over all items (and we will take $\hat{b}_0 = 1$). We now describe our strategy for computing higher-order coefficients in terms of lower-order coefficients under the assumption of *exact* queries, after which we conduct error propagation analysis. For a monomial $m$ of degree $j$, let $x_{(h,m)}$ be the point where $x_{(h,m),i} = hZ$ if an item $i$ belongs to $m$ and 0 otherwise, with higher degree terms satisfying the basis constraints (i.e. $(hZ)^3$ for a degree-3 subset of $m$, and $(hZ)^j$ for $m$), which also results in the term for a monomial containing any item not in $m$ being set to zero. Query $x_{(h,m)}$ for $2j+1$ distinct values $h$ in $\{\pm 1, \ldots, \pm(j+1)\}$. For $Z = \alpha/(2d(d+1))$ all queries lie in the $\alpha$-ball, as the $\ell_1$ norm of the positive coefficients, as well as the negative offset for item $n$, are both bounded by $\alpha/2$ in the original simplex basis. Suppose all coefficients up to degree $j-1$ are known. The result of such a query (with $z = hZ$) is equivalent to:

$$q(x_{(h,m)}) = \frac{a_m z^j + f_a(z)}{b_m z^j + f_b(z)}$$

where $f_a$ and $f_b$ are $(j-1)$-degree univariate polynomials, where each coefficient of some degree $k \leq j-1$ is expressed by summing the coefficients for degree-$k$ monomials which are subsets of $m$, for $a$ and $b$ respectively. Rearranging, we have:

$$a_m = q_i(x_{(h,m)}) \cdot b_m + \frac{q(x_{(h,m)}) \cdot f_b(z) - f_a(z)}{z^j}.$$

This gives us a linear relationship between $a_m$ and $b_m$ in terms of known quantities after just one query where $z \neq 0$. Suppose we could make *exact* queries; if we observe two distinct linear relationships, we can solve for $a_m$ and $b_m$. If each query gives us the same linear relationship, i.e. $q_i(x_{(h,m)}) = q_i(x_{(h',m)})$ for every query pair $(h, h')$, then equality also holds for each of the $(q_i(x_{(h,m)}) \cdot f_b(z) - f_a(z))/z^j$ terms. If the latter term is truly a constant function $c$:

$$\frac{q_i(x_{(h,m)}) \cdot f_b(z) - f_a(z)}{z^j} = c$$

then we also have:

$$(a_m z^j + f_a(z)) \cdot f_b(z) - (b_m z^j + f_b(z)) \cdot f_a(z) = cz^j(b_m z^j + f_b(z)).$$

Each side is a polynomial with degree at most $2j$, and thus cannot agree on $2j + 1$ points unless equality holds. However, if equality does hold, we have that either $c = 0$ or $b_m = 0$, as the left side has degree at most $2j - 1$, and both $z^j$ and $b_m z^j + f_b(z)$ are bounded away from 0 for any $z \neq 0$. If $c \neq 0$, then we have that $b_m = 0$ and $a_m = c$. If $c = 0$, then we have

$$a_m z^j f_a(z) f_b(z) - b_m z^j f_a(z) f_b(z) = 0,$$

which implies $a_m = b_m$, as $f_a(z) f_b(z)$ cannot be equal to 0 everywhere due to each $a_0^i$ and $b_0$ being positive. Our answer to $q(x_{(h,m)})$ will be bounded above 0 and below 1, allowing for us to solve for both $a_m$ and $b_m$ as

$$a_m = b_m = \frac{q_i(x_{(h,m)}) \cdot f_b(z) - f_a(z)}{(1 - q_i(x_{(h,m)}))z^j}.$$

To summarize, if given exact query answers for $2j + 1$ distinct points, we must be in one of the following cases:

- We observe at least two distinct linear relationships between $a_m$ and $b_m$ from differing query answers;

- We observe a non-zero constant $\frac{q_i(x_{(h,m)}) \cdot f_b(z) - f_a(z)}{z^j} = c$ for each query, and have $a_m = c$;

- We observe $\frac{q_i(x_{(h,m)}) \cdot f_b(z) - f_a(z)}{z^j} = 0$ for each query, and can solve for $a_m = b_m$.

To begin our error analysis for perturbed queries, we first show a bound on the size of the coefficients for a polynomial which is bounded over a range.

**Lemma 10.** *Each degree-$d'$ coefficient of $f_i^*$ is at most $d'^{2d'}$.*

*Proof.* First note that the constant coefficient and the coefficient for each linear term have magnitude at most 1, as the function is bounded in $[\lambda, 1]$ over the domain (which includes $\mathbf{0}$). For a degree-$d'$ monomial $m$, consider the univariate polynomial corresponding to moving each of its variables in synchrony while holding the remaining variables at 0, whose degree-$d'$ coefficient is equal to $a_m$. Consider the Lagrange polynomial representation of this polynomial

$$L_{d',j}(x) = \prod_{k \neq j}^{d'} \frac{x - x_k}{x_j - x_k};$$

$$\hat{f}_i(x) = \sum_{j=0}^{d'} \hat{y}_j L_{n,j}.$$

for $d' + 1$ evenly spaced points in the range $[-1/n, 1/d' - 1/n]$, which are all feasible under the simplex constraints (corresponding to $v_i \in [0, 1/d']$ in the original basis, for each $i \in m$). Each pair of points is separated by a distance of at least $1/(d'^2)$, and so the leading coefficient of each Lagrange term is at most $d'^{2(d'-1)}$. Each $\hat{y}_j$ is in $[\lambda, 1]$ and so we have

$$a_m \leq (d' + 1)d'^{2(d'-1)}$$
$$\leq d'^{2d'}$$

for each $d' > 1$. $\qquad\square$

As we estimate coefficients for monomials of increasing degree, we will maintain the invariant that each degree-$j$ coefficient of $a$ and $b$ is estimated up to additive error $\epsilon_j$, with respect to the normalization where $b_0 = 1$. Immediately we have $\epsilon_0 = \beta$ for the estimates $\hat{a}_0$ from our query to the $\mathbf{0}$ vector. We will also let $\beta_j$ denote the error of a polynomial $\hat{f}_a$ restricted to terms for subsets of a $j$-degree monomial $m$

For a monomial $m$, suppose we receive 2 queries $\hat{q}_i(x_{(h,m)})$ and $\hat{q}_i(x_{(h',m)})$ for some $h$ and $h'$ where

$$\left|\hat{q}_i(x_{(h,m)}) - \hat{q}_i(x_{(h',m)})\right| \geq F_j$$

for some quantity $F_j$. Then we have:

$$\hat{a}_m = \hat{q}_i(x_{(h,m)})\hat{b}_m + \frac{\hat{q}_i(x_{(h,m)}) \cdot \hat{f}_b(hZ) - \hat{f}_a(hZ)}{(hZ)^j}$$

$$= \hat{q}_i(x_{(h,'m)})\hat{b}_m + \frac{\hat{q}_i(x_{(h,'m)}) \cdot \hat{f}_b(h'Z) - \hat{f}_a(h'Z)}{(h'Z)^j}$$

$$\hat{b}_m = \frac{\hat{a}_m}{\hat{q}_i(x_{(h,m)})} + \frac{\frac{\hat{f}_a(hZ)}{\hat{q}_i(x_{(h,m)})} - \hat{f}_b(hZ)}{(hZ)^j};$$

$$= \frac{\hat{a}_m}{\hat{q}_i(x_{(h',m)})} + \frac{\frac{\hat{f}_a(h'Z)}{\hat{q}_i(x_{(h',m)})} - \hat{f}_b(h'Z)}{(h'Z)^j};$$

$$\frac{\hat{a}_m}{\hat{q}_i(x_{(h',m)})} - \frac{\hat{a}_m}{\hat{q}_i(x_{(h,m)})} = \frac{\frac{\hat{f}_a(hZ)}{\hat{q}_i(x_{(h,m)})} - \hat{f}_b(hZ)}{(hZ)^j} - \frac{\frac{\hat{f}_a(h'Z)}{\hat{q}_i(x_{(h',m)})} - \hat{f}_b(h'Z)}{(h'Z)^j};$$

$$\hat{a}_m = \frac{\frac{\hat{q}_i(x_{(h',m)})\hat{f}_a(hZ)}{\hat{q}_i(x_{(h,m)})} - \hat{q}_i(x_{(h',m)})\hat{f}_b(hZ)}{\left(1 - \frac{\hat{q}_i(x_{(h',m)})}{\hat{q}_i(x_{(h,m)})}\right) \cdot (hZ)^j} - \frac{\hat{f}_a(h'Z) - \frac{\hat{f}_b(h'Z)}{\hat{q}_i(x_{(h',m)})}}{\left(1 - \frac{\hat{q}_i(x_{(h',m)})}{\hat{q}_i(x_{(h,m)})}\right) \cdot (h'Z)^j};$$

$$\hat{b}_m = \frac{\frac{\hat{q}_i(x_{(h,m)}) \cdot \hat{f}_b(hZ) - \hat{f}_a(hZ)}{(hZ)^j} - \frac{\hat{q}_i(x_{(h,'m)}) \cdot \hat{f}_b(h'Z) - \hat{f}_a(h'Z)}{(h'Z)^j}}{\hat{q}_i(x_{(h',m)}) - \hat{q}_i(x_{(h',m)})};$$

where $\hat{f}_a$ and $\hat{f}_b$ are the univariate polynomials from summing the lower-order coefficient estimates for each degree up to $j-1$. The additive error to each $\hat{f}_a(hZ)$ and $\hat{f}_a(hZ)$ can be bounded by:

$$\beta + \sum_{k=1}^{j-1}\binom{n}{k}(hZ)^k k^{2k}\epsilon_k = \beta + \sum_{k=1}^{j-1}\binom{n-1}{k}(k^2 hZ)^k\epsilon_k.$$

Further, the magnitude of each $\hat{f}_a(hZ)$ and $\hat{f}_b(hZ)$ is at most $1 + \sum_{k=1}^{j-1}\binom{n-1}{k}(k^2 hZ)^k$. We can bound the error of other terms as follows:

- Each $\hat{q}_i(x_{(h',m)}) - \hat{q}_i(x_{(h',m)})$ has magnitude at least $F_j$ and at most 1, and additive error at most $2\beta$;

- Each $\hat{q}_i(x_{(h',m)})$ has value at least $\frac{\lambda}{n}$ and at most 1, and additive error at most $\beta$;

- Each $\frac{\hat{q}_i(x_{(h',m)})}{\hat{q}_i(x_{(h,m)})}$ term is either greater than $\frac{1}{1-F_j}$ or at most $1 - F_j$; the true ratio between the numerator and denominator is at least $\lambda/n$ most $n/\lambda$, with additive error up to $\beta$ in both.

- Each $1 - \frac{\hat{q}_i(x_{(h',m)})}{\hat{q}_i(x_{(h,m)})}$ term, is either greater than $F_j$ or at most $1 - \frac{1}{1-F_j}$;

- Each $(hZ)^j$ has magnitude at least $Z^j$;

The error in the numerator of $\hat{a}_m$, and the fractional terms in the numerator of $\hat{b}_m$ is dominated by multiplying the functions of $\hat{q}_i$ with the polynomials themselves. As such, we can bound the error to

$a_m$ and $b_m$ by $\epsilon_j$ if we have that:

$$\epsilon_j \geq O\left(\frac{n\beta}{\lambda F_j Z^j} \cdot \left(1 + \sum_{k=1}^{j-1} \binom{n-1}{k}(k^2 hZ)^k\right)\right)$$

$$= O\left(\frac{n\beta}{\lambda F_j Z^j} \cdot \left(1 + \sum_{k=1}^{j-1} \binom{n-1}{k}(h\alpha)^k\right)\right)$$

$$= O\left(\frac{nd^{2j}\beta}{\lambda \alpha^j F_j}\right)$$

for any $\alpha < 1/(nd)$. Now suppose all pairs of query answers we see are separated by less than $F_j$; the additive error to each estimate of the quantity

$$\hat{c}_{(h,m)} = \frac{\hat{q}_i(x_{(h,m)}) \cdot \hat{f}_b(z) - f_a(z)}{(hZ)^j}$$

is $\mathcal{E}_j = O\left(\frac{\beta}{Z^j} \cdot \left(1 + \sum_{k=1}^{j-1}\binom{n}{k}(k^2 hZ)^k\right)\right) = O\left(\beta \cdot nd^{2j}/\alpha^j\right)$. If each such quantity has value at most $\mathcal{E}_j$, we assume this quantity is zero and solve for $a_m = b_m$. If some are larger, we must be in the case where $\hat{b}_m \approx 0$ and so we set $a_m = \hat{c}_{(h,m)}$ for any query result. By taking each $F_j = O(\sqrt{\beta}\,\text{poly}(n, d^j, 1/\alpha^j))$ we can obtain a bound of $\epsilon_j = O(\sqrt{\beta}\,\text{poly}(n, d^j, 1/\alpha^j))$ to each coefficient regardless of which case we are in; after summing the error contribution across coefficients and accounting for renormalization, recalling that $\lambda = \Omega(1/n)$, we obtain a bound of $\epsilon$ on score vector errors (for any desired norm) provided that $\epsilon \geq \sqrt{\beta}\,\text{poly}(n, d^d, 1/\alpha^d)$. $\qquad\square$

Next, we prove the local learnability result for normalized multivariate polynomials.

**Lemma 11.** $\mathcal{M}_{BMNP}$ *is $O(n^d)$-locally learnable by an algorithm $\mathcal{A}_{BMNP}$ with $\beta \leq \frac{\epsilon}{\alpha^d F(n,d)}$, where $F(n,d)$ is some function depending only on $n$ and $d$ which is finite for all $n, d \in \mathbb{Z}$.*

*Proof.* Our approach will be to construct a set of $O(n^d)$ queries which results in a data matrix which is nonsingular in the space of $d$-degree multivariate polynomials, solve for the coefficients of each $f_i$ as a linear function over this basis, and show that the basis is sufficiently well-conditioned such that our approximation error is bounded.

Consider the set of polynomials where each $v_n$ term is reparameterized as $1 - \sum_{i=1}^{n-1} v_i$, then translated so that the uniform vector appears at the origin (i.e. with $v_n = -\sum_{i=1}^{n-1} v_i$). Our approach will be to learn a representation of each polynomial directly, as they are already normalized to sum to a constant (which must be in the range $[1, n]$). Let $f_i^*$ be the representation of $f_i$ in this translation. Let $\mathcal{B}$ be the $N = \sum_{j=0}^{d}(n-1)^j$-dimensional basis where each variable in a vector $x$ corresponds to a monomial of variables in $v$ with degree at most $d$, with the domain constrained to ensure mutual consistency between monomials, e.g.:

$$\mathcal{B} = \{1, v_1, \ldots, v_{n-1}, v_1^2, v_1 v_2, \ldots, v_{n-1}^d\}.$$

Observe that $f_i^*$ is a linear function in this basis, with $f_i^*(x) = \langle a, x \rangle$ and $\sum_{i=1}^{n} f_i^*(x) = \langle b, x \rangle$ for any $x$ represented in $\mathcal{B}$.

There is a large literature on constructing explicit query sets for multivariate polynomial interpolation, which ensure that the resulting data matrix is nonsingular; see [14] for an overview. The set must have at least $N$ points to ensure uniqueness of interpolation, and this is sufficient when points are appropriately chosen. Let $S^*$ be any such set such that each point $\|w\|_1 \leq 1/2$ for each $w \in S^*$, and let $C_{n,d}$ be the $\ell_\infty$ condition number of the resulting matrix $Y$ (which will be positive due to nonsingularity) given by:

$$Y = \begin{bmatrix} y_1^{(1)} & \cdots & y_N^{(1)} \\ \vdots & & \vdots \\ y_1^{(j)} & \cdots & y_N^{(j)} \\ \vdots & & \vdots \\ y_1^{(N)} & \cdots & y_N^{(N)} \end{bmatrix}$$

where $y^{(j)}$ is the representation of $s^{(j)}$ in the basis $\mathcal{B}$. We show that for any $\alpha$, we can construct a matrix $X$ from a query set $S^\alpha$ of size $N$ where $\|v\|_1 \leq \alpha/2$ for each $v \in S^\alpha$. For each $s^{(j)}$, let $v^{(j)} = \alpha s^{(j)}$, which results in $\|v\|_1 \leq \alpha/2$ for the parameterization over $n-1$ items, and so radius of $\alpha$ holds when including all $n$ items. This results in a matrix $X$ given by

$$
X = \begin{bmatrix}
x_1^{(1)} & \cdots & x_N^{(1)} \\
\vdots & & \vdots \\
x_1^{(j)} & \cdots & x_N^{(j)} \\
\vdots & & \vdots \\
x_1^{(N)} & \cdots & x_N^{(N)}
\end{bmatrix}
$$

We then have

$$X = YD,$$

where $D$ is a diagonal matrix with the $j$th diagonal entry $\nu_j$ equal to $\alpha^{d_j}$, where $d_j$ is degree of the $j$th monomial in $\mathcal{B}$, as our scaling by $\alpha$ is amplified for each column in correspondence with the associated degree; the values of $D$ will range from $\alpha^d$ to 1. We can then bound the condition number of $X$ as:

$$
\begin{aligned}
\operatorname{cond}(X) &= \operatorname{cond}(YD) \\
&= \|YD\| \, \|(DY)^{-1}\| \\
&\leq \|Y\| \, \|D\| \, \|D^{-1}\| \, \|Y^{-1}\| \\
&= \operatorname{cond}(Y) \cdot \operatorname{cond}(D) \\
&\leq C_{n,d} \frac{\max_j \nu_j}{\min_j \nu_k} \\
&= \frac{C_{n,d}}{\alpha^d}.
\end{aligned}
$$

Let $q$ denote the vector of exact answers to each query in $x$ from $f_i$, equal to $a\dot{x}$ and let $\hat{q}$ be the answers we observe for item $i$ from querying each $x$. As $X$ is nonstationary, we have that $Xa = q$, and by standard results in perturbation theory for linear systems, for $\hat{a}$ such that $X\hat{a} = \hat{q}$ we have that:

$$
\begin{aligned}
\frac{\|\hat{a} - a\|}{\|a\|} &\leq \operatorname{cond}(X) \frac{\|\hat{q} - q\|}{\|q\|} \\
&\leq \frac{\beta n C_{n,d}}{k^2 \alpha^d}
\end{aligned}
$$

as each entry in $q$ is at least $\lambda \geq k^2/n$. Further note that the maximum coefficient of a degree-$d$ multivariate polynomial which takes maximum value 1 over the unit ball (and hence the simplex) can be shown to be bounded by a finite function of $n$ and $d$ (see [19]); when accounting for this factor in relative error across all terms and items, as well as the condition number, we have that for $\beta \leq \frac{\epsilon}{\alpha^d F(n,d)}$ for some function $F(n,d)$, the scores generated by the functions $\hat{f}_i$ using our estimated coefficients $\hat{a}$ result in score vector estimates bounded by $\epsilon$.

$\square$

## C.3 Proof of SFR Local Learnability

We now prove that functions with local sparse Fourier transformation are locally learnable. Recall that a function $f(x)$ has a $\ell$-sparse Fourier transform if it can be written as

$$
f(x) = \sum_{i=1}^{\ell} \xi_i e^{2\pi \mathbf{i} \eta_i x},
$$

where $\eta_i$ is the $i$-th frequency, $\xi_i$ is the corresponding magnitude, and $\mathbf{i} = \sqrt{-1}$.

We will use the following result about learning sparse Fourier transforms [25].

**Theorem 6** ([25]). *Consider any function $f(x) : \mathbb{R} \to \mathbb{R}$ of the form*

$$f(x) = f^*(x) + g(x) \,,$$

*where $f^*(x) = \sum_{i=1}^{\ell} \xi_i e^{2\pi \mathbf{i} \eta_i x}$ with frequencies $\eta_i \in [-F, F]$ and frequency separation $\hat{\alpha} = \min_{i \neq j} |\eta_i - \eta_j|$, and $g(x)$ is the arbitrary noise function. For some parameter $\delta > 0$, we define the noise-level over an interval $I = [a, b] \subseteq \mathbb{R}$ as*

$$\mathcal{N}^2 = \frac{1}{|I|} \int_I |g(x)|^2 dx + \delta \sum_{i=1}^{\ell} |\xi_i|^2 \,.$$

*There exists an algorithm that takes samples from the interval $I$ with length $|I| > O(\frac{\log(\ell/\delta)}{\hat{\alpha}})$ and returns a set of $\ell$ pairs $\{(\xi_i', \eta_i')\}$ such that for any $|\xi_i| = \Omega(\mathcal{N})$ we have for an appropriate permutation of the indices*

$$|\eta_i - \eta_i'| = O\Big(\frac{\mathcal{N}}{|I||\xi_i|}\Big), \qquad |\xi_i - \xi_i'| = O(\mathcal{N}), \forall i \in [\ell] \,.$$

*The algorithm takes $O(\ell \log(F|I|) \log(\frac{\ell}{\delta}) \log(\ell))$ samples and $O(\ell \log(F|I|) \log(\frac{F|I|}{\delta}) \log(\ell))$ and succeeds with probability at least $1 - 1/k^c$ for any arbitrary constant c.*

Furthermore, the algorithm used in the above theorem uses samples of the form $x_0, x_0 + \sigma \cdots x_0 + \ell \log(\ell/\delta)\sigma$ for randomly chosen $x_0$ and $\sigma = O(|I|/\ell \log(\ell/\delta))$.

We will use the above theorem to learn the sparse Fourier representation of the preference model. Recall that for a memory vector $v$ and item $i \in [n]$, $M(v)_i = f_i(v_i)$.

*Proof.* Let $v_{\text{unif}}$ denote the uniform memory vector. We will learn each function $f_i$ separately. Fix $i \in [n]$. We will set the interval $I$ to be $[1/n - Z, 1/n + Z]$ for some sufficiently small $\frac{\log(\ell/\delta)}{\hat{\alpha}} \leq Z \leq \alpha/2$ where $\hat{\alpha}$ is the frequency separation, where $\alpha = \tilde{\Omega}(1/\hat{\alpha})$ so that $Z$ is defined. Let $S = \{x_j\}_{j=1}^{\tilde{O}(\ell)}$ for $x_j \in [-Z, Z]$ be a set of points such that the Fourier learning algorithm queries $1/n + x$ for each $x \in S$. For each point $x \in S$, we define the memory vector $v^x = v_{\text{unif}} + xe_i - xe_j$ where $j$ is a fixed randomly chosen other index. All such vectors lie in $V_\alpha$, as $2(\alpha/2)^2 \leq \alpha^2$. We query all vectors $v^x$ for $x \in S$, along with $v_{\text{unif}}$. Recall that $\hat{s}_v$ is the empirical score vector at a memory vector $v$. For each vector $v$, let $R_v$ be the sum of all scores of all the $n - 2$ items held at $\frac{1}{n}$, divided by the sum of those same items' scores in the uniform vector $v_{\text{unif}}$. For each vector $v^x$ we multiply the score $\hat{s}_{v^x, i}$ of item $i$ by $R_{v^x}$ to obtain a noisy sample of $f_i(1/n + x)$. For $i \in \tilde{O}(\ell)$, let the $i$-th sample be denoted by $\hat{y}_i$ and the true value $f_i(1/n + x_i)$ be denoted by $y_i$. We then pass all these samples to the Fourier learning algorithm in Theorem 6 in order to get an estimate $\hat{f}$ of $f$.

We now analyze the error in the samples. Let $R_v^*$ be the corresponding ratio of these sums of scores under $\{f_i\}$; each sum is within $[\frac{\lambda}{3}, 1]$ at each vector, and the sums of observed scores have additive error at most $2n\beta$. As such, $R_v$ has additive error at most $\frac{2n\beta}{\lambda}$ from $R_v^*$. For each vector $v^x$ we have that $\hat{s}_{v^x, i}/(\sum_j \hat{s}_{v^x, j})$ is within a $\beta$ error from $s_{v^x, i}/(\sum_j s_{v^x, j})$. Hence, the total error in each sample is bounded as:

$$|\hat{y}_i - y_i| \leq \frac{7n\beta}{\lambda} \,.$$

Using this we can bound the total noise term by $\mathcal{N} = 8n\beta/\lambda$ using our choice of $\delta = (\beta n)/(\lambda \sum_{i=1}^{\ell} |\xi_i|)$. The algorithm will return a set of $\{(\hat{\eta}_i, \hat{\xi}_i)\}$ such that

$$|\eta_i - \eta_i'| = O\Big(\frac{1}{\alpha}\Big), \qquad |\xi_i - \xi_i'| = O(\frac{\beta n}{\lambda}), \forall i \in [\ell] \,.$$

So for function $\hat{f}_i(x)$ we can bound its distance from $f_i(x)$ by:

$$\left| f_i(x) - \hat{f}_i(x) \right| = \left| \sum_{i=1}^{\ell} \xi_i e^{2\pi \mathbf{i} \eta_i x} - \sum_{i \in [\ell]} \hat{\xi}_i e^{2\pi \mathbf{i} \hat{\eta}_i x} \right|$$

$$\leq \sum_{i \in [\ell]} \left| \xi_i e^{2\pi \mathbf{i} \eta_i x} - \hat{\xi}_i e^{2\pi \mathbf{i} \hat{\eta}_i x} \right|$$

$$\leq \sum_{i \in [\ell]} \left| \xi_i - \hat{\xi}_i \right| |\eta_i - \hat{\eta}_i|$$

$$\leq O(\frac{\ell n \beta}{\lambda \alpha}),$$

since we normalize the above estimates to get a score estimate, the total bound on the score estimates can be bounded as:

$$\left| \frac{\hat{f}_i(x)}{\sum_{j=1}^{x} \hat{f}_j(x)} - \frac{f_i(x)}{\sum_{j=1}^{x} f_j(x)} \right| \leq O(\frac{\ell \beta n}{\alpha \lambda}).$$

Taking $\beta \leq \frac{\epsilon \lambda \alpha}{\sqrt{n} \ell}$ gives us an error of at most $\epsilon \sqrt{n}$, satisfying a Euclidean bound of $\epsilon$ from any true score vector $M(w)/M_w^*$ for our hypothesis model $\hat{M}(v) = \{\hat{f}_i(v_i) : i \in [n]\}$. □

# D   Omitted Proofs for Section 3

## D.1   Proof of Theorem 1

*Proof.* First observe that $y_t \in \mathcal{K}_t$ every round, as

For $x^* = \arg\min_{x \in \mathcal{K}_T} \sum_{t=1}^T f_t(x)$, let $x_{\delta,\epsilon}^* = \Pi_{\mathcal{K}_{T,\delta,\epsilon}}(x^*)$. By linearity and properties of projection, we also have that $x_{\delta,\epsilon}^* = \arg\min_{x \in \mathcal{K}_{T,\delta,\epsilon}} \sum_{t=1}^T f_t(x)$, and that $\left\| x_{\delta,\epsilon}^* - x^* \right\| \leq (\delta + \epsilon)\frac{D}{r}$. For $G$-Lipschitz losses $\{f_t\}$ we have

$$\sum_{t=1}^T \mathbb{E}[\phi_t] - \sum_{t=1}^T f_t(x^*) = \sum_{t=1}^T \mathbb{E}[f_t(y_t)] - \sum_{t=1}^T f_t(x^*)$$

$$\leq \sum_{t=1}^T \mathbb{E}[f_t(y_t)] - \sum_{t=1}^T f_t(x_{\delta,\epsilon}^*) + \delta TG\frac{D}{r} + \epsilon TG\frac{D}{r}.$$

Let $\hat{f}_t(x) = \mathbb{E}_{u \sim \mathbb{B}}[f(x + \delta u + \xi_t)] = f_t(x + \xi_t)$ by linearity. Then we can bound the regret by:

$$\sum_{t=1}^T \mathbb{E}[\phi_t] - \sum_{t=1}^T f_t(x^*) \leq \sum_{t=1}^T \mathbb{E}[f_t(y_t)] - \sum_{t=1}^T f_t(x_{\delta,\epsilon}^*) + \frac{\delta TGD}{r} + \frac{\epsilon TGD}{r}$$

$$= \sum_{t=1}^T \mathbb{E}[\hat{f}_t(x_t)] - \sum_{t=1}^T f_t(x_{\delta,\epsilon}^*) + \frac{\delta TGD}{r} + \frac{\epsilon TGD}{r}$$

$$\leq \sum_{t=1}^T \mathbb{E}[\hat{f}_t(x_t)] - \sum_{t=1}^T \hat{f}_t(x_{\delta,\epsilon}^*) + \frac{\delta TGD}{r} + \epsilon TG\left(\frac{D}{r} + 1\right)$$

$$\leq \sum_{t=1}^T \mathbb{E}[\hat{f}_t(x_t)] - \sum_{t=1}^T \hat{f}_t(x_{\delta,\epsilon}^*) + \frac{\delta TGD}{r} + \frac{2\epsilon TGD}{r}$$

Next, we prove a series of lemmas — an analysis of online gradient descent for contracting decision sets, and a corresponding bandit-to-full-information reduction — which allow us to view the remaining summation terms involving $\{x_t\}$ as the expected regret of stochastic online gradient descent for the loss function sequence $\{\hat{f}_t\}$ with respect to $\mathcal{K}_{T,\delta,\epsilon}$.

When modifying online gradient descent to project into smaller sets each round, the analysis is essentially unchanged.

---

**Algorithm 3** Contracting Online Gradient Descent.

---

Input: sequence of contracting convex decision sets $\mathcal{K}_1, \ldots \mathcal{K}_T$, $x_1 \in \mathcal{K}_1$, step size $\eta$
Set $x_1 = \mathbf{0}$
**for** $t = 1$ to $T$ **do**
    Play $x_t$ and observe cost $f_t(x_t)$
    Update and project:

$$y_{t+1} = x_t - \eta \nabla \ell_t(x_t)$$
$$x_{t+1} = \Pi_{\mathcal{K}_{t+1}}(y_{t+1})$$

**end for**

---

**Lemma 12.** *For a sequence of contracting convex decision sets $\mathcal{K}_1, \ldots \mathcal{K}_T$, $x_1 \in \mathcal{K}_1$ each with diameter at most $D$, a sequence of $G$-Lipschitz losses $\ell_1, \ldots, \ell_T$, and $\eta = \frac{D}{G\sqrt{T}}$, the regret of Algorithm 3 with respect to $\mathcal{K}_t$ is bounded by*

$$\sum_{t=1}^{T} \ell_t(x_t) - \min_{x^* \in \mathcal{K}_T} \sum_{t=1}^{T} \ell_t(x^*) \leq GD\sqrt{T}.$$

*Proof.* Let $x^* = \arg \min_{x \in \mathcal{K}_T} \sum_{t=1}^{T} \ell_t(x)$, and let $\nabla_t = \nabla \ell_t(x_t)$. First, note that

$$\ell_t(x_t) - \ell_t(x^*) \leq \nabla_t^\top (x_t - x^*)$$

by convexity; we can then upper-bound each point's distance from $x^*$ by:

$$\|x_{t+1} - x^*\| = \left\| \Pi_{\mathcal{K}_{t+1}}(x_t - \eta \nabla \ell_t(x_t)) \right\| \leq \|x_t - \eta \nabla_t - x^*\|,$$

using projection properties for convex bodies. Then we have

$$\|x_{t+1} - x^*\|^2 \leq \|x_t - x^*\|^2 + \eta^2 \|\nabla_t\|^2 - 2\eta \nabla_t^\top (x_t - x^*)$$

and

$$\nabla_t^\top (x_t - x^*) \leq \frac{\|x_t - x^*\|^2 - \|x_{t+1} - x^*\|^2}{2\eta} + \frac{\eta \|\nabla_t\|^2}{2}.$$

We can then conclude:

$$
\begin{aligned}
\sum_{t=1}^{T} \ell_t(x_t) - \sum_{t=1}^{T} \ell_t(x^*) &\leq \sum_{t=1}^{T} \nabla_t^\top (x_t - x^*) \\
&\leq \sum_{t=1}^{T} \frac{\|x_t - x^*\|^2 - \|x_{t+1} - x^*\|^2}{2\eta} + \frac{\eta}{2} \sum_{t=1}^{T} \|\nabla_t\|^2 \\
&\leq \frac{\|x_T - x^*\|^2}{2\eta} + \frac{\eta}{2} \|\nabla_t\|^2 \\
&\leq \frac{D^2}{2\eta} + \frac{\eta}{2} \sum_{t=1}^{T} \|\nabla_t\|^2 \\
&= GD\sqrt{T} \qquad\qquad\qquad \left(\text{when } \eta = \frac{D}{G\sqrt{T}}\right)
\end{aligned}
$$

$\square$

The bandit-to-full-information reduction is fairly standard as well, with a proof equivalent to that of e.g. Lemma 6.5 in [17], modified for a full-information algorithm $\mathcal{A}$ for over contracting sets.

**Lemma 13.** *Let $u$ be a fixed point in $\mathcal{K}_T$, let $\{\ell_t : \mathcal{K}_t \to \mathbb{R} \mid t \in [T]\}$ be a sequence of differentiable loss functions, and let $\mathcal{A}$ be a first-order online algorithm that ensures a regret bound $Regret_{\mathcal{K}_T}(\mathcal{A}) \leq B_{\mathcal{A}}(\nabla \ell_1(x_1), \ldots, \nabla \ell_T(x_T))$ in the full-information setting for contracting sets $\mathcal{K}_1, \ldots, \mathcal{K}_T$. Define the points $\{x_t\}$ as $x_1 \leftarrow \mathcal{A}(\emptyset)$, $x_t \leftarrow \mathcal{A}(g_1, \ldots, g_{t-1})$, where $g_t$ is a random vector satisfying*

$$\mathbb{E}[g_t | x_1, \ell_1, \ldots, x_t, \ell_t] = \nabla \ell_t(x_t).$$

*Then for all $u \in \mathcal{K}_T$:*

$$\mathbb{E}[\sum_{t=1}^{T} \ell_t(x_t)] - \sum_{t=1}^{T} \ell_t(u) \leq E[B_{\mathcal{A}}(g_1, \ldots, g_T)] \tag{1}$$

*Proof.* Let $h_t : \mathcal{K}_t \to \mathbb{R}$ be given by:

$$h_t(x) = \ell_t(x) + \psi_t^\top x, \text{ where } \psi_t = g_t - \nabla \ell_t(x_t).$$

Note that $\nabla h_t(x_t) = g_t$, and so deterministically applying a first order algorithm $\mathcal{A}$ on $\{h_t\}$ is equivalent to applying $\mathcal{A}$ on stochastic first order approximations of $\{f_t\}$. Thus,

$$\sum_{t=1}^{T} h_t(x_t) - \sum_{t=1}^{T} h_t(u) = \ \leq B_{\mathcal{A}}(g_1, \ldots, g_T).$$

Using the fact that the expectation of each $\psi_t$ is 0 conditioned on history, and expanding, we get that

$$
\begin{aligned}
\mathbb{E}[h_t(x_t)] &= \mathbb{E}[\ell_t(x_t)] + \mathbb{E}[\psi_t^\top x_t] \\
&= \mathbb{E}[\ell_t(x_t)] + \mathbb{E}[\mathbb{E}[\psi_t^\top x_t | x_1, \ell_1, \ldots, x_t, \ell_t]] \\
&= \mathbb{E}[\ell_t(x_t)] + \mathbb{E}[\mathbb{E}[\psi_t | x_1, \ell_1, \ldots, x_t, \ell_t]^\top x_t] \\
&= \mathbb{E}[\ell_t(x_t)],
\end{aligned}
$$

and we can conclude by taking the expectation of Equation 1 for any point $u \in \mathcal{K}_T$. $\qquad \square$

The key remaining step is to observe that each $g_t$ is an unbiased estimator of $\nabla \hat{f}_t(x_t)$:

$$
\begin{aligned}
\mathbb{E}[g_t | x_1, \hat{f}_1, \ldots, x_t, \hat{f}_t] &= \frac{n}{\delta} \mathbb{E}[\phi_t u_t | x_t, \hat{f}_t] \\
&= \frac{n}{\delta} \mathbb{E}[\mathbb{E}[\phi_t | x_t, \hat{f}_t, u_t] \cdot u_t | x_t, \hat{f}_t] \\
&= \mathbb{E}[f_t(x_t + \delta u_t + \xi_t) u_t | x_t, \hat{f}_t] \\
&= \mathbb{E}[\hat{f}_t(x_t + \delta u_t) u_t] \\
&= \nabla \hat{f}_t(x_t),
\end{aligned}
$$

where the final line makes use the sphere sampling estimator for linear functions (as in e.g. Lemma 6.7 in [17]). This allows us to apply Lemma 13 to Algorithm 3:

$$
\begin{aligned}
\sum_{t=1}^{T} \mathbb{E}[\phi_t] - \sum_{t=1}^{T} f_t(x^*) &\leq \sum_{t=1}^{T} \mathbb{E}[\hat{f}_t(x_t)] - \sum_{t=1}^{T} \hat{f}_t(x_{\delta,\epsilon}^*) + \frac{\delta T G D}{r} + \frac{2\epsilon T G D}{r} \\
&\leq \text{Regret}_{COGD}\left(g_1, \ldots, g_T | \{\hat{f}_t\}\right) + \frac{\delta T G D}{r} + \frac{2\epsilon T G D}{r} \\
&\leq \frac{D^2}{2\eta} + \frac{\eta}{2} \sum_{t=1}^{T} \|g_t\|^2 + \frac{\delta T G D}{r} + \frac{2\epsilon T G D}{r} \\
&\leq \frac{D^2}{2\eta} + \eta \frac{n^2}{2\delta^2} T + \frac{\delta T G D}{r} + \frac{2\epsilon T G D}{r} \qquad (\text{def. of } g_t, \phi \leq 1) \\
&\leq n G D T^{3/4} + \frac{G D T^{3/4}}{r} + \frac{2\epsilon T G D}{r} \qquad (\eta = \frac{D}{n T^{3/4}}, \delta = \frac{1}{T^{1/4}}).
\end{aligned}
$$

$\qquad \square$

# E   Omitted Proofs for Section 4

## E.1   Proof of Lemma 4

*Proof.* Consider any memory vector $v \in \Delta(n)$. We can show constructively that there is some distribution of menus $z_U$ which induces the all-$\frac{1}{n}$ vector.

We construct $z_U$ in $\frac{1}{\tau} + 1$ stages for some $\tau > 0$, through a process where we continuously add weight $a_{z_j}$ to a sequence of distributions $\{z_j | j \geq 1\}$ over menus until the total weight $\sum_j a_{z_j}$ sums to 1. The uniform-inducing menu distribution $z_U$ will then be defined by taking the mixture of the menu distributions $z_j$ where each is weighted by $a_{z_j}$.

Consider the uniform distribution over all menus; continuously add weight to this distribution until some item (the one with the largest score in $M$) has selection weight $\tau/n$ (its selection probability under $M$ at memory vector $v$ in each distribution of menus $z_j$ considered thus far, weighted by $a_{z_j}$). While there are at least $k$ items with selection weight $\tau/n$, continuously add weight to the uniform distribution over all menus containing only items with weight below $\tau/n$.

Once there are fewer than $k$ items with selection weight at most $\tau/n$, we terminate stage 1. In general, for stage $i$, we always include every item with weight below $\tau i/n$ in the menu, with all others chosen uniformly at random.

Inductively, we can see that every item starts stage $i$ with at least weight $\tau(i-1)/n$ and at most $\tau i/n$, with at most $k-1$ items having weight less than $\tau(i-1)/n$. Crucially, any item with weight less than $\tau i/n$ at the start of stage $i$ will reach weight $\tau i/n$ before any item starting at weight $\tau i/n$ reaches weight $\tau(i+1)/n$. Such an item is included in every menu until this occurs, resulting in a selection probability of at least $\frac{\lambda}{k}$ in each menu distribution considered, whereas any other item is only included in the menu with probability at most $\frac{k}{n}$, which bounds its selection probability in the menu distribution. As $\frac{\lambda}{k} \geq \frac{k}{n}$, the selection weight of items beginning stage $i$ below $\tau i/n$ reaches $\tau i/n$ no later than when the stage terminates.

After stage $\frac{1}{\tau}$, every item has weight at most $\frac{1}{n}$ and at least $\frac{1}{n} - \frac{\tau}{n}$. We continue for one final stage until the sum of weights is 1, at which point every item has a final weight $p_{z_U} \in [\frac{1}{n} - \frac{\tau}{n}, \frac{1}{n} + \frac{\tau}{n}]$. Taking the limit of $\tau$ to zero gives us that $x_U$ is in $\mathtt{IRD}(v, M)$ for any $v$, and hence $x_U$ is in $\mathtt{EIRD}(M)$ as well.

Further, there is a distribution of menus $z_{b_i}$ where $i$ has probability $p_{b_i, i} = k/n$ and every other item $j$ has probability

$$p_{b_i, j} = \frac{1}{n} - \frac{k-1}{n(n-1)}$$

Here, we include $i$ in every menu and run the previous approach over the remaining $n-1$ items for menus of size $k-1$, which we then augment with $i$. The required bound on $\lambda$ still holds for any $\lambda < 1$, as $\frac{k^2}{n} \geq \frac{(k-1)^2}{n-1}$ (for any $k \leq \sqrt{n}$, which holds as $\lambda < 1$). The selection probability of $i$ will be at least $\frac{\lambda}{k} \geq \frac{k}{n}$; we can take a mixture of this menu distribution with $z_U$ such that $p_{b_i, i} = \frac{k}{n}$ exactly.

The convex hull of each $p_{b_i}$ is thus contained in $\mathtt{EIRD}(M)$, as any point $p \in \mathrm{convhull}\{p_{b_i} | i \in [n]\}$ can be generated by taking the corresponding convex combination of menu distributions $z_{b_i}$. Any point $x \in \Delta(n)$ where $\|x_U - x\|_\infty \leq \frac{k-1}{n(n-1)}$ can then be induced by taking mixtures of the $z_{b_i}$ menu distributions. $\qquad\square$

## E.2   Subset-Uniform Distributions in $\mathtt{EIRD}$

**Lemma 14.** *For any $\lambda$-dispersed $M$ where $\lambda \geq \frac{Ck^2}{n}$, $\mathtt{EIRD}(M)$ contains the uniform distribution over any $\frac{n}{C}$ items.*

*Proof.* The proof of Lemma 4 carries through directly for a universe with only $\frac{n}{C}$ items. $\qquad\square$

### E.3 Implementing Near-Uniform Vectors

**Lemma 15.** *For any $\lambda$-dispersed $M$ where $\lambda \geq \frac{k^2}{n}$, for any point $x \in \Delta(N)$ satisfying*

$$\|x - x_U\|_\infty \leq \frac{k-1}{n(n-1)},$$

*there is an adaptive strategy for selecting a sequence of menus over $t^*$ rounds, resulting in a $t^*$-round empirical distribution $\hat{x}$ such that $\|x - \hat{x}\|_\infty \leq \gamma t^* + O(n)$ with probability at least $1 - 2n \exp(-\gamma^2 t^*/8)$, for any $\gamma$.*

*Proof.* Our strategy will essentially correspond to the construction in Lemma 4, which shows that our vector is indeed in $\mathrm{EIRD}(M)$. For each item $i$, let $V_i = t^* \cdot x_i$ be the target number of rounds where $i$ is selected over the window. For any $t \leq t^*$ let $\hat{V}_{t,i}$ be the number of additional rounds an item must be selected before reaching its target, with $\hat{V}_{1,i} = V_i$. In each round $t$, construct a menu for the Agent by choosing the $k$ items with largest remaining counts $\hat{V}_{t,i}$, breaking ties uniformly at random, and decrement by 1 the count of the item selected in that round. Our approach will be to show that each item's final count under this process is close to its target in expectation after $t^*$ rounds, and use the sequence of expectations as rounds progress to define a martingale which will be close to its final expectation with high probability.

Let $\hat{V}_{t,\perp}$ denote the minimum value of $\hat{V}_{t,i}$ across items. Observe that our procedure maintains the invariant that $\hat{V}_{t,\perp}$ can only decrease in a round where at most $k - 1$ items have remaining counts $\hat{V}_{t,i} > \hat{V}_{t,\perp}$. We will consider each round in which $\hat{V}_{t,\perp}$ decreases as the beginning of a "trial", and we will track the expectations of $\hat{V}_{t,i}$ over sequences of trials across two cases:

- Case 1: For every round $t$ at the start of a trial, we have had $\hat{V}_{t,i} - \hat{V}_{t,\perp} > 2$;

- Case 2: There has been some round $t$ at the start of a trial where $\hat{V}_{t,i} - \hat{V}_{t,\perp} \leq 2$.

When the first trial begins, we have at most $k - 1$ items in Case 1, and items can never enter Case 1 after being in Case 2. We assume without loss of generality that we begin in a state where the first trial has just begun, as no prior rounds can increase the distance of any item from the minimum.

*Case 1.* Note that the probability of an item in the menu being selected in a given round is at least $\lambda/k \geq k/n$. We can upper-bound the expected distance of some count $\hat{V}_{t,i}$ from $\hat{V}_{t,\perp}$ by analyzing a "pessimistic" process where we assume that this minimum selection probability is tight, where every selection of an item other than item $i$ corresponds to the beginning of an "event", where the number of selections of $i$ in each event is geometrically distributed with parameter $p = 1 - \frac{k}{n}$. While these counts are not truly geometrically distributed, as the maximum number of selections is bounded, we will only need to analyze the probabilities of sums corresponding to items remaining in Case 1, in which case truncation does not affect the resulting distribution. Not every event corresponds to a new trial; there are deterministically at least $n - k$ events per trial, as every item begins a trial with a strictly higher count than $\hat{V}_{t,\perp}$, and so at least $n - k - 1$ selections of items other than $i$ must occur before an item with minimum count can enter the menu (conditioned on $\hat{V}_{t,i}$ remaining above $\hat{V}_{t,\perp}$).

Under this process, after $z$ events, the distribution of $\hat{V}_{t,i}$ is given by subtracting the sum of $z$ of the aforementioned geometric variables from $\hat{V}_{1,i}$, which is distributed according to a negative binomial:

$$\Pr\left[\hat{V}_{1,i} - \hat{V}_{t,i} = y\right] = \binom{z + y - 1}{z - 1} \left(\frac{k}{n}\right)^y \left(1 - \frac{k}{n}\right)^z$$

with mean $\frac{z(k/n)}{1-k/n} = \frac{zk}{n-k} = \mathbb{E}[y]$ and variance $\frac{zk/n}{(1-k/n^2)}$. After $z$ events, $\hat{V}_{1,\perp}$ has dropped by at most $\frac{z}{n-k}$. As such, by the time $\hat{V}_{t,\perp}$ reaches 0, we would also have that the expectation of $\hat{V}_{i,t}$ would reach 0 if we were to keep item $i$ in the menu at every round and allowed its count to drop below $\hat{V}_{t,\perp}$ without replacing it (and become negative); however, our process truncates (and enters Case 2) upon reaching 2 from the minimum, and so we can simply show that the contribution of the left tail of this distribution is small. Note that at the beginning of our process, we have $\hat{V}_{1,i} - \hat{V}_{1,\perp} \leq \frac{t^*(k-1)}{n(n-1)}$, and

so the expected difference from the minimum upon reaching $\hat{V}_{t,\perp} = 0$ while remaining in Case 1 is at most:

$$\mathbb{E}[(\hat{V}_{t^*,i} - \hat{V}_{t^*,\perp}) \cdot I(\text{Case } 1)] \leq 2 + \sum_{y=0}^{\hat{V}_{1,i}-2} (2 + \hat{V}_{1,i} - y)\binom{t^*-1}{t^*-y-1}\left(\frac{k}{n}\right)^y\left(1 - \frac{k}{n}\right)^{t^*-y}$$

$$\leq 2 + \sum_{y=0}^{\hat{V}_{1,i}-2} \frac{(2 + \hat{V}_{1,i} - y)t^*}{t^* - \hat{V}_{1,i}}\binom{t^*}{t^*-y}\left(\frac{k}{n}\right)^y\left(1 - \frac{k}{n}\right)^{t^*-y}.$$

For any $y$ in this range we have going from $y-1$ to $y$:

$$\frac{\binom{t^*}{t^*-y}\left(\frac{k}{n}\right)^y\left(1-\frac{k}{n}\right)^{t^*-y}}{\binom{t^*}{t^*-y-1}\left(\frac{k}{n}\right)^{y-1}\left(1-\frac{k}{n}\right)^{t^*-y-1}} = \frac{t^*-y-1}{y}\cdot\frac{\frac{k}{n}}{1-\frac{k}{n}}$$

$$\geq \frac{t^*}{\hat{V}_{1,i}}\cdot\frac{k}{n}$$

$$\geq \frac{t^*}{1/n + k/n^2}\cdot\frac{k}{n}$$

$$\geq \frac{k}{1 + k/n}$$

which is greater than 1 for any $k \geq 2$. As such, we can bound the tail summation by:

$$\mathbb{E}[(\hat{V}_{t^*,i} - \hat{V}_{t,\perp}) \cdot I(\text{Case } 1)] \leq 2 + \frac{t^*}{t^* - \hat{V}_{1,i}}\cdot\sum_{y=0}^{\infty}\left(\frac{1+k/n}{k}\right)^y$$

$$\leq 2 + \frac{t^*}{t^* - \hat{V}_{1,i}}\cdot\frac{k}{k-1-k/n}$$

$$\leq 5$$

for $k \geq 2$ and sufficiently large $n$.

*Case 2.* Here we show that once an item has reached Case 2, its expected distance from $\hat{V}_{t,\perp}$ in any future round is at most a constant. Separating this analysis is necessitated by the fact that there exist edge cases where an item's expected distance from the minimum can be increasing (e.g. if all items start a trial at one above the minimum, an item can only have a decreasing distance if it becomes the next minimum, and can have a higher likelihood of remaining in the menu when the next trial begins). Our approach will be to show by induction that, beginning from the first trial in Case 2, the distribution of item $i$'s distance from the minimum, where $p_y$ is the probability of distance $y$, satisfies:
$$p_{y+1} \leq p_y/2^{k/2-1}$$
for $y \geq 2$. This holds at the first trial in Case 2, as we have $p_{y+1} = 0$ for each $y \geq 2$. An item can only have a distance increase of 1 in a given trial (if it is not picked in any of the at least $n-k$ rounds), which occurs with probability at most $\frac{1}{(1+k/n)}^{n-k} \leq e^{-k/2} \leq \frac{1}{2^{k/2}}$, using that $n-k > n/2$ (which holds given that $k \geq 2$ and $n \geq k^2$). Further, using the same negative binomial process as in Case 1 to describe the number of selections of item $i$ in a given trial, we can see that $1/2$ upper bounds its density function after $n-k$ events for any valid setting of our parameters, and so the probability that an item is selected $j$ times, for $j$ such that it remains in every menu, is at most $1/2$. Letting $p^*$ describe the distribution after another trial, we can solve for:

$$p_y^* = p_{y-1}/2^{k/2} + \sum_{j=0}^{\infty} p_{y+j}\cdot\Pr[\text{drops } j+1]$$

$$\leq p_{y-1}/2^{k/2} + p_y/2;$$
$$p_{y+1}^* = zp_y/2^{k/2-1};$$

using the induction hypothesis on $p$. As such, in any future trial, the expected distance from minimum can be given by:

$$\mathbb{E}[\hat{V}_{t^*,i} - \hat{V}_{t,\perp}|\text{Case 2}] \leq 2 + \sum_{y=3}^{\infty} p_y$$

$$\leq 2 + \sum_{y=3}^{\infty} 2^{y(k/2-1)}$$

$$\leq 3.21$$

for any $k \geq 3$. One can strengthen this to yield a constant sum for $k = 2$ via a more delicate analysis on the upper bound of the negative binomial density function, which we omit.

*Concentration Analysis.* We now have that in either case, the expectation $\mathbb{E}[(\hat{V}_{t^*,i} - \hat{V}_{t^*,\perp})]$ is a constant, for every item $i$. Given any current empirical counts counts $\{\hat{V}_{t,i} : i \in [n]\}$ and scores for every item at any time $t$ (which we as the Recommender need not know), the distribution over subsequent items chosen is fully defined. Let $X_{t,i} = \Pr[\, i$ chosen $\mid \{\hat{V}_{t-1,i} : i \in [n]\}, \{f_i(v_t)\}]$. For this process, we can now view each quantity $Y_{t,i} = (\hat{V}_{t-1,i} - \hat{V}_{t,i})$ as a Bernoulli random variable with mean $X_{t,i}$. Then we can define $Z_{t,i} = \sum_{h=1}^{t} Y_{h,i} - X_{h,i}$ as a martingale, where $\mathbb{E}[Z_{t,i}] = Z_{t-1,i}$ and $|Z_{t,i} - Z_{t-1,i}| \leq 2$. Note that $\mathbb{E}[Z_{t^*,i}]$ is equal to $V_i$ up to a small constant $c_i$. We can then apply Azuma's inequality to get:

$$Pr\left[\left|\hat{Z}_{t^*,i} - V_i - c_i\right| \geq \gamma t^*\right] \leq 2\exp\left(\frac{-\gamma^2 t^*}{8}\right).$$

These constants are independent of $t^*$, and will vanish when $t^*$ is sufficiently large. $\qquad\square$

### E.4 Proof of Theorem 5

*Proof.* Let:

- $F_{LL} = f_{LL}(\lambda, \alpha, n, \mathcal{M})$ s.t. $\mathcal{A}_{\mathcal{M}}$ with $\beta/F_{LL}$ results in $\epsilon_{LL} = \frac{\epsilon\lambda k}{n}$;

- $F_Q = \frac{8L\sqrt{n}k}{\lambda}F_{LL}$;

- $t_{\text{query}} = \frac{2n}{k-1}\left(\frac{F_{LL}}{\beta}\right)^2 \log\left(\frac{2nkS}{(k-1)\delta_{\text{query}}}\right) = \tilde{\Theta}(1/\epsilon^2)$;

- $t_{\text{pad}} = \max\left(\frac{2F_Q t_{\text{query}}}{\beta}, \frac{32n^2 F_Q^2 \log(2/\delta_{\text{pad}})}{\beta^2}\right) = \tilde{\Theta}(1/\epsilon^3)$;

- $t_{\text{move}} = \max\left(\frac{n(n-1)t_{\text{query}}}{k-1}, \frac{32n^2 F_Q^2 \log(4S/\delta_{\text{move}})}{(1-4k/n)\beta^2}, t_{\text{pad}}\right) = \tilde{\Theta}(1/\epsilon^3)$;

- $t_0 = t_{\text{pad}} + S(2 \cdot t_{\text{move}} + t_{\text{query}}) = \tilde{\Theta}(1/\epsilon^3)$.

After running `UniformPad` via the first Lemma 15 construction for $t_{\text{pad}}$ steps, our empirical memory vector is within $\ell_\infty$ distance $\frac{\beta}{nF_Q}$ of $x_U$ with probability at least $1 - \delta_{\text{pad}}$. We maintain the invariant that when calling `MoveTo(x)` to reach some non-uniform vector $x$ from $x_U$, the $\ell_\infty$ distance between $x$ and $x_U$ is at most $\alpha$, and that after calling `Query(x)` the current vector $x'$ (accounting for drift during sampling) has $\ell_\infty$ distance at most $\alpha$ from $x_U$.

At any time $t < t_0$ when `MoveTo` is called, the proportion of steps which the current invocation will contribute to the total history is at least:

$$R_{\text{move}} = \frac{t_{\text{move}}}{t_{\text{pad}} + S(t_{\text{move}} + t_{\text{query}})} = O(1/S)$$

Let $\alpha = \frac{k-1}{2n(n-1)} \cdot R_{\text{move}}$ denote the radius of the $\ell_2$ ball around $x_U$ in which we permit queries for local learning. Any point $x$ within the $\alpha$-ball around the uniform vector can reach (or be reached

from) the uniform vector with one call to `MoveTo`$(x)$, as their $\ell_\infty$ distance is at most $\alpha$, so some difference vector exists with mass $R_{\text{move}}$ and which satisfies the required norm bound. For each input $x$, called from $x_t$, `MoveTo`$(x)$ applies the construction from Lemma 15 for the mass $t_{\text{move}}$ vector $y = x \cdot (t_{\text{move}}) - x_t \cdot t$. This results in a total error of at most $\frac{\beta}{2nF_Q} \cdot t_{\text{move}} + 1 \leq \frac{\beta}{nF_Q} \cdot t_{\text{move}}$ per item count with probability at least $1 - \delta_{\text{move}}$, as

$$t_{\text{move}} \geq \frac{32n^2 F_Q^2 \log(4S/\delta_{\text{move}})}{(1 - 4k/n)\beta^2}.$$

This yields a total variation distance within $\frac{\beta}{2F_Q}$ for the entire memory vector when appended to the current history.

To run `Query`$(x)$, consider a set of $\frac{n}{k-1}$ menus, where item 1 appears in every menu and every other item appears in exactly one. Over the following $t_{\text{query}}$ rounds, play each menu $t_{\text{query}} \cdot \frac{k-1}{n}$ times and note the proportion of each item observed relative to item 1 when its menu was played. Each scoring function $f_i \in M$ is $L$-Lipschitz; we run `Query`$(x)$ for $t_{\text{query}}$ rounds, which can introduce a drift of at most $\beta/(2F_Q)$ in total variation distance given the bound on $t_{\text{query}}$ in terms of $t_{\text{pad}}$. This drift results in a vector which remains within $\ell_\infty$ distance $2\alpha$ from $x_U$, and so $x_U$ can still be reached again in a single `MoveTo`$(x_U)$ call.

The empirical average memory vector over all menu queries (for any item) is within $\beta/F_Q$ total variation distance from $x$, and so the expected distribution of items differs from that at $x$ by at most $\beta/F_Q \cdot \frac{4L\sqrt{nk}}{\lambda} = \beta/(2F_{LL})$ in $\ell_\infty$ distance. Each point's observed frequency differs from that expectation by at most $\beta/(2F_{LL})$ with high probability. For an item $i$ in the menu at a given round, we view whether or not it was chosen as a Bernoulli random variable, with mean equal to its relative score among items in the menu. Let $\bar{s}_{v,K,i}$ be the expected frequency of observing an item when the menu $K$ containing it is played, given the empirical sequence of memory vectors during those rounds $t_{\text{query}} \cdot \frac{k-1}{n}$, and let $\hat{s}_{v,K,i}$ be the true observed frequency. We then have:

$$Pr\left[|\bar{s}_{v,K,i} - \hat{s}_{v,K,i}| \geq \frac{\beta}{2F_{LL}}\right] \leq 2e^{\left(-2(\beta/2F_{LL})^2 t_{\text{query}}(k-1)/n\right)}$$

$$= 2e^{\left(-(\beta/F_{LL})^2 t_{\text{query}}(k-1)/(2n)\right)}$$

$$\leq \frac{\delta_{\text{query}}(k-1)}{nkS},$$

given that

$$t_{\text{query}} \geq \frac{2n}{k-1}\left(\frac{F_{LL}}{\beta}\right)^2 \log\left(\frac{2nkS}{(k-1)\delta_{\text{query}}}\right).$$

For item 1 take the average over all menus, and rescale such that all scores sum to 1 (using the frequency of item $i$ relative to the frequency of item 1 when both were in the menu). Each score, and its error bound, will only shrink under the rescaling. This gives us score vector estimates $\hat{s}_x$ for each $x \in S$ with additive error at most $\frac{\beta}{F_{LL}}$ relative to the true frequency of item 1, and thus overall, where $F_{LL} = f_{LL}(\lambda, \alpha, n, \mathcal{M})$. This holds for every query simultaneously with probability $1 - \delta_{\text{query}}$.

By the local learnability guarantee for $\mathcal{M}$, running $\mathcal{A}_{\mathcal{M}}$ our results in a hypothesis $\hat{M}$ which has $\ell_2$ error at most $\epsilon_{LL} = \frac{\epsilon\lambda k}{n}$ for any $x \in \Delta(n)$. In each round, the model and memory vector defines a space of feasible item distributions. This allows us to run RC-FKM for perturbations up to $\epsilon$. We can represent each set `IRD`$(v_t, \hat{M})$ explicitly as the convex hull of normalized score estimates for every menu.

We implement `PlayDist`$(x)$ using current score estimates $\hat{M}(v_t)$ to generate a menu distribution which approximately induces the instantaneous item distribution $x$. Taking the convex hull over every menu's score vector under $\hat{M}$ yields a polytope representation of `IRD`$(v_t, \hat{M})$, which will contain our chosen action at each step.

**Lemma 16.** *Let $x$ be a point in `IRD`$(v, M)$, and let $z \in \Delta(\binom{n}{k})$ be a non-negative vector such that $\sum_{j \in \binom{n}{k}} z_j \cdot p_{K_j, v} = x$, where $K_j$ is the jth menu in lexicographic order. If the Recommender randomly selects a menu $K$ to show the Agent with probability according to $z$, then the Agent's item selection distribution is $x$.*

*Proof.* The probability that the Agent selects item $i$ is obtained by first sampling a menu, then selecting an item proportionally to its score:

$$\Pr[\text{Agent selects } i] = \sum_{j \in \binom{n}{k}} z_j \cdot p_{K_j,v,i} = x_i.$$

$\square$

**Lemma 17.** *Given $\hat{M}$ satisfying $\frac{\epsilon \lambda k}{n}$-accuracy and a target vector $x_t \in \text{IRD}(v_t, \hat{M})$ generated by RC-FKM, there is a linear program for computing a menu distribution $z_t$ such that the induced item distribution $p_{z_t}$ satisfies*

$$\|p_{z_t} - x_t\| \le \epsilon.$$

*Proof.* We can define a linear program to solve for $z$ with:

- variables for $z_j \in [0,1]$, where $\sum_{j \in \binom{n}{k}} z_j = 1$,

- estimated induced distributions for each menu $\hat{p}_{K_j}$, and

- a constraint for each $i \in [n]$:

$$\sum_{j=1}^{\binom{n}{k}} z_j \cdot \hat{p}_{K_j,i} = x_{t,i}.$$

If $\left\| \hat{M}(x)/\hat{M}^* - M(x)/M_x^* \right\| \le \frac{\epsilon \lambda k}{n}$, then for any menu distribution $z$, we have that:

$$\|p_{z,v} - \hat{p}_{z,v}\| \le \epsilon.$$

Consider some menu $K$. The $\ell_2$ distance of score vectors restricted to the menu is at most $\frac{\epsilon \lambda k}{n}$, and each vector has mass at least $\frac{k\lambda}{n}$ by dispersion. Rescaling vectors to have mass 1 yields a bound of $\epsilon$, which is preserved under mixture (which is the induced distribution by Lemma 16), as well as when projecting into the $n-1$ dimensional space for RC-FKM, and so there is some perturbation vector $\xi_t$ with norm at most $\epsilon$ such that $z$ induces $x_t + \xi_t$. $\square$

Note that the losses for RC-FKM can be $2G$-Lipschitz after the reparameterization where $x_{t,n} = 1 - \sum_{i=1}^{n-1} x_{t,i}$. Any point satisfying within radius $r = \frac{k-1}{n(n-1)}$ from the uniform distribution in $n$ dimensions, feasible by Lemma 4, is within distance $r$ under the reparameterization as well, as we simply drop the term for $x_n$. The required radius surrounding $\mathbf{0}$ for RC-FKM of $r$ is thus satisfied, and we have that $\epsilon + \delta \le r/T^{1/4} \le r$. Further, the diameter of the simplex is bounded by $D = 2$. We can directly apply the regret bound of RC-FKM for these quantities, which holds with respect to $H_c \cap \text{EIRD}(\hat{M})$. By Lemma 17, for any point $x \in \text{EIRD}(\hat{M})$, there is a point $x' \in \text{EIRD}(M)$ such that $\|x - x'\| \le \epsilon$. Projecting both points into $H_c$ cannot increase their distance by convexity, and so the optimality gap between the two sets is at most $\epsilon GT$. Our total regret is at most the sum of:

- Maximal regret for the learning runtime $G \cdot t_0$;

- The regret of RC-FKM over $T - t_0$ rounds;

- The gap between $\text{EIRD}(\hat{M})$ and $\text{EIRD}(M)$; and

- The union bound of each event's failure probability.

We can bound this by:

$$\text{Regret}_{C \cap \text{EIRD}(M)}(T) \le G \cdot t_0 + 4nGT^{3/4} + \frac{4(\delta + 2\epsilon)GT}{r} + \epsilon GT + (\delta_{\text{pad}} + \delta_{\text{move}} + \delta_{\text{query}})T$$

$$= \tilde{O}(T^{3/4})$$

when taking each of $\{\delta_{\text{pad}}, \delta_{\text{move}}, \delta_{\text{query}}\} = \frac{1}{T^{1/4}}$. We can also bound the empirical distance from $H_c$.

**Lemma 18.** *The diversity constraint is $O(\epsilon)$-satisfied by the empirical distribution $v_T$ with probability $1 - O(T^{-1/4})$.*

*Proof.* Note that after $t_0$, the empirical distribution $v_{t_0}$ is within total variation distance $\frac{\beta}{2F_Q}$ from $x_U$ (which is necessarily in $H_c$). Further, each vector $x_t$ played by RC-FKM results in a per-round expected item distribution $y_t$ which lies in $H_c$ by the robustness guarantee. We can apply a similar martingale analysis as in Lemma 15 to the sequence of realizations of any item versus its cumulative expectation $\sum_{t > t^*} y_t$ to get a bound of (much less than) $\frac{\beta}{2F_Q}$ in total variation distance as well, which is preserved under mixture. For any locally learnable class, $\beta = O(\epsilon)$. Note that for all the classes we consider, we have $\beta/(2F_Q) \ll \epsilon$. Both events hold with probability $1 - O(T^{-1/4})$, as we can apply the same failure probabilities used for the learning stage for each.

Note that for a constraint $H_c$ where $c$ is sufficiently bounded away from $\log(n)$ and for large enough $T$, this will in fact yield an empirical distribution which exactly satisfies $H_c$, as the weight $\tilde{O}(T^{3/4})$ uniform window will "draw" the empirical distribution back towards the center of $H_c$, as it dominates the total $\tilde{O}(T^{1/2})$ total error bound (for the unnormalized empirical histogram $T \cdot v_T$) obtainable with a martingale analysis over the entire RC-FKM window. $\square$

This completes the proof of the theorem.

$\square$