# OpenReview forum: "Diversified Recommendations for Agents with Adaptive Preferences"
_NeurIPS.cc/2022/Conference — NeurIPS 2022 Accept_

### Official Review · Reviewer_9Hsd · 2022-07-10

**Rating:** 7
**Confidence:** 1
**Soundness:** 3 good
**Presentation:** 4 excellent
**Contribution:** 3 good

**Summary:**

This paper studies a game in which menus of items are shown to a user.  The user has unknown preferences which influence the choice of the item from the menu (exfactly one item is chosen).

The chosen item has two consequences:
1) it modifies the agents future behavior and makes it more likely to select the same action
2) it gives the recommender a reward which is non-personalized and depends only on the chosen item.



**Questions:**


I think the presentation would be improved by examples / case studies and possibly simulation studies as well.

**Limitations:**

The authors contribution to me is a type of bandit problem rather than taylored to RecSys.  Producing a more convincing RecSys motivation would be interesting.  Although perhaps the paper will stand on its own as a "bandit" paper - I will leave that for others to judge.

The term "adversarial" is used.  I take this to mean something "zero sum" or game theoretic is going on.  I don't see why this would occur, usually the recommender system typically has incentives to improve the user experience.

**Strengths And Weaknesses:**


The problem of feedback loops in recommender systems is interesting and there is a lack of good papers studying this question so it was a pleasure to see this paper address the topic.

That said this problem is not a clear analogue of a recommendation problem I know.  I expect people who work on industrial Recsys might have difficulty seeing the connection to the problem being studied and real Recsys problems, although this may not be the intent.

The complete formulation of a recommender system in a bandit framework contains user behavior interspersed with (usually) slates of recommendations and response to the recommendations.  Reward as measured at A/B test time is computed over the time line and may include clicks but also sales or engagement.  The intervention that an A/B test measures is therefore a combinatorial object and 'diversity' is measured within the slates and within time.  Reward itself is not attached to the recommendations but to the timeline.  It is entirely possible that a diverse collection of recommendations result in receiving reward, but this can only be understood if we attach the reward to the timeline not to the individual recommendations.  This paper like most of the recsys literature attaches the reward to the individual recommendations.

Diversity across different user timelines is also necessary for exploration purposes.

This paper makes interesting observations in a simplified model where there are causal implications in the timeline of a recommendation.  Although it isn't to my mind diversity in any of the above senses.  Indeed if I understand correctly only the selected action gives a reward and this is a non-personalized reward although the selected reward has long term impacts on the user (which is very novel).

The writing is quite dense for a recommender systems audience right now.  I would suggest augmenting the mathematical presentation with examples to increase accessibility.

---

> ### Author Response · Authors · 2022-08-02
> **Response to Reviewer9Hsd**
>
> Thank you for your helpful comments!
>
>
> ### Theoretical Motivations and Experiments:
> Our primary goal for this paper is to introduce and analyze a stylized model for considering recommendation feedback loops in a theoretically rigorous manner, through the lens of regret minimization, to better understand the inherent limitations and avoidable pitfalls of dynamic recommendation tasks. To our knowledge, this is the first regret minimization paper with item menus for an agent which accounts for such feedback loops; for example, the feedback in "dueling bandits" papers is decoupled from the history in prior rounds.
> Implementing and empirically evaluating online recommendation algorithms for various models of adaptive agents is certainly an important direction for future research.
> We will add some exposition which grounds the theory more directly in examples.
>
> ### Single-Agent Restrictions:
> Our model aims to capture specific phenomena relating to online recommendations, and is not intended to represent the full complexity of an applied recommender systems setting. We view our work as a step in the direction of representing more of this complexity via a bandit approach; our model is related to that in "dueling bandits" papers, which feature menus of recommendations for a single agent but do not incorporate adaptive preferences.
> In particular, we leave the aggregation of preferences across multiple users for future work. The single-agent setting is already quite rich when accounting for adaptive preferences. Deciding how to incorporate multiple agents in this setting is certainly an interesting question, and determining the appropriate model is not entirely straightforward.
>
> ### "Naive" Multi-Agent Extensions:
> While not practically efficient, one could imagine a Recommender using an separate instance of the algorithm in parallel for every Agent on the platform, and every Agent may have a distinct preference model and associated set of reward functions. Additionally, if Agent "types" correspond to their preference model but each Agent maintains a distinct history, we could use an instance of the algorithm for each type and speed up local learning by "batching" queries across Agents; after local learning is complete, the same regret bound could be obtained for the RC-FKM window, as we know each Agent's type and history. A more satisfying multi-agent model would likely involve a more elegant way of correlating preferences across users than entirely distinct preference models for discrete Agent types, which we are exploring for future work.
>
> ### Rewards:
> As for rewards, we  assume that the Recommender can estimate the reward resulting from the menu of recommendations at time $t$ after observing the Agent's selection and before showing a new menu of recommendations at time $t+1$.
> These rewards can indeed be personalized by Agent, and can vary arbitrarily over time.
> We only need an unbiased estimate of the reward resulting from that interaction, as the inner RC-FKM algorithm is robust to stochastic feedback. For example, if a user watches a recommended video and clicks on the ad, they may or may not purchase that product at a later date (resulting in more reward), but knowing their conditional probability of a purchase given a click is sufficient. Our algorithm maximizes this average expected reward with respect to the EIRD set.
>
> In general, some amount of reward decomposition into individual time steps is necessary to a obtain a sublinear regret bound. Our lower bounds show that we can't even optimize over the set of fixed menus, which is a simpler problem than optimizing over the set of sequences of recommendations.
>
>
> ### Diversity vs Entropy:
> By "diversity", we mean long-run sequences of selections which are not too concentrated on a small number of items. We note in Section 2 that, with the exception of Theorem 2, our positive results hold for any constraint set which is convex and contains the uniform distribution; these are natural requirements for any definition of a diversity constraint, as the uniform distribution is "maximally diverse" and taking a mixture of "sufficiently diverse" distributions should result in a sufficiently diverse distribution. We will make this clearer in the camera-ready version. We emphasize entropy because it is well-studied as a measure of dispersion for discrete distributions, as well as for its connection to the EIRD set that we show in Theorem 2.

---

> > ### Author Response · Authors · 2022-08-02
> > **Response, continued:**
> >
> >
> > ### Adversarial Rewards:
> > For no-regret learning, the term "adversarial rewards" is in contrast to fixed or stochastic losses, meaning that the loss functions can vary arbitrarily over time (as if chosen by an "adversary" who wants to trick you into obtaining high regret, which can be viewed as a zero-sum game). A Recommender wants to maximize their resulting reward, which is a function of both user experience and advertiser interest. Handling adversarial losses enables us to hedge against the possibility that the most profitable items to recommend on a given day may change unpredictably. For example, some world event may cause a drop in advertiser interest for certain content, or a new group of advertisers may emerge who are all interested in advertising alongside content that was previously not as profitable for the Recommender. Further, there could be cyclical patterns in advertiser interest; a company placing winter coat ads on skiing videos won't want to spend as much in summer months, so we should avoid bombarding a user exclusively with these videos (even if they're happy to watch year round). The "stickiness" of preferences is another such motivation for diverse recommendations, which our results emphasize.

---

### Official Review · Reviewer_e35G · 2022-07-10

**Rating:** 7
**Confidence:** 4
**Soundness:** 3 good
**Presentation:** 3 good
**Contribution:** 3 good

**Summary:**

The paper discusses the problem of online recommendation for a single agent with adaptive preferences as an adversarial bandit problem. The authors show that it is necessary to consider a different benchmark, as there are no linear regret algorithms that can compete against the typical best menu/item distribution. Since the available bandit optimization algorithms do not apply to contracting decision sets, the authors provide a new algorithm for this setting that also handles imprecision from estimation. Under a class of locally learnable functions and a mild assumption on each preference model, the authors supply a local learning algorithm to learn an estimated model that achieves $\tilde{O}(T^{3/4})$ regret.

**Questions:**

- How can this model be applied to the setting with a single recommender and $N$ different agents? Could the current model handle having more agents whose preferences and behaviors are correlated?

- Could there be experimental sections that show the performance of the second algorithm? While the discussions of the subroutines in section 4.2 are helpful, it could be useful seeing how the algorithm performs on some synthetic data.

**Limitations:**

- The authors sufficiently address the limitations in Section 5.

- While the authors address the negative societal impact of having a small set of items recommended, more discussion on the societal effect of having diversified recommendations could be useful in section 1.

**Strengths And Weaknesses:**

Strengths:
- The paper investigates an interesting problem of recommendation with adaptive preferences and provides novel theoretical results in their lower bounds.

- The theoretical contributions are significant in advancing the study of diversification in recommendation problems.

- The authors provide a comprehensive set of assumptions and sufficient explanations for why they are necessary.

- The first algorithm (RC-FKM) is intuitive, and the second algorithm has sufficient discussion.

- The authors adequately address the current literature in their related work section and the appendix.

- The paper is well-organized and easy to follow.

Weaknesses:

- The current model only considers a single agent, which is restrictive.

- In section 2, when the authors first introduce the diversity constraint, the variable $d_{TV}$ was not defined before (line 152).

- In section 3, algorithm 1 and theorem 4 lack sufficient discussion of the results.

- The theoretical results could benefit from having a small proof sketch next to them.

- In line 261, there is an extra "and" at the end.

---

> ### Author Response · Authors · 2022-08-02
> **Response to Reviewer e35G**
>
>
> Thank you for your helpful comments!
>
>
> ### Theoretical Motivations and Experiments:
> Our primary goal for this paper is to introduce and analyze a stylized model for considering recommendation feedback loops in a theoretically rigorous manner, through the lens of regret minimization, to better understand the inherent limitations and avoidable pitfalls of dynamic recommendation tasks. To our knowledge, this is the first regret minimization paper with item menus for an agent which accounts for such feedback loops; for example, the feedback in "dueling bandits" papers is decoupled from the history in prior rounds.
> Implementing and empirically evaluating online recommendation algorithms for various models of adaptive agents is certainly an important direction for future research.
>
> ### Single-Agent Restrictions:
> Our model aims to capture specific phenomena relating to online recommendations, and is not intended to represent the full complexity of an applied recommender systems setting. We view our work as a step in the direction of representing more of this complexity via a bandit approach; our model is related to that in "dueling bandits" papers, which feature menus of recommendations for a single agent but do not incorporate adaptive preferences.
> In particular, we leave the aggregation of preferences across multiple users for future work. The single-agent setting is already quite rich when accounting for adaptive preferences. Deciding how to incorporate multiple agents in this setting is certainly an interesting question, and determining the appropriate model is not entirely straightforward.
>
> ### "Naive" Multi-Agent Extensions:
> While not practically efficient, one could imagine a Recommender using an separate instance of the algorithm in parallel for every Agent on the platform, and every Agent may have a distinct preference model and associated set of reward functions. Additionally, if Agent "types" correspond to their preference model but each Agent maintains a distinct history, we could use an instance of the algorithm for each type and speed up local learning by "batching" queries across Agents; after local learning is complete, the same regret bound could be obtained for the RC-FKM window, as we know each Agent's type and history. A more satisfying multi-agent model would likely involve a more elegant way of correlating preferences across users than entirely distinct preference models for discrete Agent types, which we are exploring for future work.
>
> ### Notation and Impacts:
> $d_{TV}$ refers to the total variation distance, which we will state explicitly for the camera-ready version. We will add more discussion for the intuition behind RC-FKM in Section 3, as well as sketches of the proofs throughout. We will add some discussion of further societal impacts to the conclusion (e.g. pros and cons of discouraging "rabbit holes").

---

> > ### Comment · Reviewer_e35G · 2022-08-08
> > **Response to the authors**
> >
> > Thank you for the rebuttal. I have read the rebuttal and my opinion has not changed.

---

### Official Review · Reviewer_DAFz · 2022-07-11

**Rating:** 5
**Confidence:** 1
**Soundness:** 2 fair
**Presentation:** 2 fair
**Contribution:** 3 good

**Summary:**

The paper 'Diversified Recommendations for Agents with Adaptive Preferences' looks at the problem of recommending a set of items in an online setting where customer preferences adapt over time to satisfy both the diversified recommendations criteria as well as to avoid an effect of the "rabbit hole". The authors formulate the problem as an adversarial bandit task and demonstrate a negative result in terms of finding an algorithm that satisfies all the proposed constraints. The authors then suggest an algorithm for bandit linear optimisation.

**Questions:**

* in the abstract (line 23) you mention that you give a set of negative results justifying your assumptions. Do you refer to Theorem 1?
* It would be nice to add references to the appendix, like in the case of the Theorem 1 proof.
* while I do not mind reading papers with unconventional structure, I found in this case that hiding the literature overview in the appendix is making it even more difficult to read.
* there is an interesting claim that the further users go down the rabbit hole, the more likely they are to continue. While I agree, I think it would be good to substantiate this claim. (line 31)
* (lines 60-61) you write that you propose a 'natural benchmark'. I'm not sure I understand what it means in this context.
* (lines 108, 126) typos
* (line 152) I couldn't find where $d_{TV}$ is introduced

**Limitations:**

Yes

**Strengths And Weaknesses:**

My review of strengths and weaknesses will be relatively brief. As I'm quite inexperienced with the topic, I admit that it was very difficult to read and understand. The paper poses a very interesting problem (high originality) and conducts several theoretical analyses to provide insights into the problem. As the intended audience must be much more familiar with the concepts, I can't comment on this part, but I might recommend to incorporate empirical results or clarifying simulations/visualisations based on the suggested Algorithm 2 or RC-FKM. This might help to grasp the results and the outlined concepts for the more inexperienced audience. Therefore, I will omit the quality and significance from my review and would rate clarity on a medium-to-low level.

---

> ### Author Response · Authors · 2022-08-02
> **Response to Reviewer Reviewer DAFz**
>
> Thank you for your helpful comments!
>
> ### Theoretical Motivations and Experiments:
> Our primary goal for this paper is to introduce and analyze a stylized model for considering recommendation feedback loops in a theoretically rigorous manner, through the lens of regret minimization, to better understand the inherent limitations and avoidable pitfalls of dynamic recommendation tasks. To our knowledge, this is the first regret minimization paper with item menus for an agent which accounts for such feedback loops; for example, the feedback in "dueling bandits" papers is decoupled from the history in prior rounds.
> Implementing and empirically evaluating online recommendation algorithms for various models of adaptive agents is certainly an important direction for future research.
>
> ### Single-Agent Restrictions:
> Our model aims to capture specific phenomena relating to online recommendations, and is not intended to represent the full complexity of an applied recommender systems setting. We view our work as a step in the direction of representing more of this complexity via a bandit approach; our model is related to that in "dueling bandits" papers, which feature menus of recommendations for a single agent but do not incorporate adaptive preferences.
> In particular, we leave the aggregation of preferences across multiple users for future work. The single-agent setting is already quite rich when accounting for adaptive preferences. Deciding how to incorporate multiple agents in this setting is certainly an interesting question, and determining the appropriate model is not entirely straightforward.
>
> ### "Naive" Multi-Agent Extensions:
> While not practically efficient, one could imagine a Recommender using an separate instance of the algorithm in parallel for every Agent on the platform, and every Agent may have a distinct preference model and associated set of reward functions. Additionally, if Agent "types" correspond to their preference model but each Agent maintains a distinct history, we could use an instance of the algorithm for each type and speed up local learning by "batching" queries across Agents; after local learning is complete, the same regret bound could be obtained for the RC-FKM window, as we know each Agent's type and history. A more satisfying multi-agent model would likely involve a more elegant way of correlating preferences across users than entirely distinct preference models for discrete Agent types, which we are exploring for future work.
>
> ### Audience:
> The target reader for the paper is indeed someone who is familiar with online convex optimization, bandits, or online learning theory more broadly. We will add exposition to the camera-ready version to emphasize the kinds of results a reader with a different background might want to consult first, as well as intuition for RC-FKM (in short: it functions similarly to stochastic gradient descent, where a gradient estimator is constructed at each step, and the average loss over all rounds is minimized).
>
> ### "Natural Benchmarks":
> For regret minimization tasks, a "natural benchmark" is one which is attainable but nontrivial; for many problems this is simply the best item in the action set. For our setting, we show that analogues of that "standard benchmark" are unattainable (item or menu distributions), but that EIRD already captures the "distributions of interest" when high diversity is desired (Theorem 2), and additionally is feasible.
>
> ### Other Comments:
> $d_{TV}$ refers to the total variation distance, which we will state explicitly for the camera-ready version (and address the mentioned typos).
> The "negative results" refer to Theorems 1 and 3, for justifying the EIRD and local learnability restrictions, respectively.
> We will add appendix references for the camera-ready version, as well as expand the literature review in Section 1.3. The papers we reference in the sentences following the "rabbit hole" discussion do indeed provide empirical evidence for these phenomena, but we will also make this more clear.

---

### Official Review · Reviewer_sqQK · 2022-07-11

**Rating:** 5
**Confidence:** 3
**Soundness:** 3 good
**Presentation:** 2 fair
**Contribution:** 3 good

**Summary:**

The paper formalizes the recommendation process as a bandit task. The agent recommends items based on a preference model of the user learning to maximize rewards based on a diversity constraint.  They also introduce locally learnable functions that can be learned from a small neighborhood i.e., behavior over the entire domain can be learned from a small region



**Questions:**

Some notations are used before being defined. What is s_(v,i)?, d_TV

How do you correlate entropy and diversity?


**Limitations:**

I was unable to find discussions on limitations

**Strengths And Weaknesses:**

Strengths:
The paper gives insights on the importance of diversity in recommendations
The paper also has proofs in formalizing a benchmark for regret
Locally learnable functions are defined fir learning from a small neighborhood


Weakness
Though there are proofs showcasing the results, there is no evaluation on real data to support the claims of the paper.

---

> ### Author Response · Authors · 2022-08-02
> **Response to Reviewer sqQK**
>
> Thank you for your helpful comments!
>
> ### Theoretical Motivations and Experiments:
> Our primary goal for this paper is to introduce and analyze a stylized model for considering recommendation feedback loops in a theoretically rigorous manner, through the lens of regret minimization, to better understand the inherent limitations and avoidable pitfalls of dynamic recommendation tasks. To our knowledge, this is the first regret minimization paper with item menus for an agent which accounts for such feedback loops; for example, the feedback in "dueling bandits" papers is decoupled from the history in prior rounds.
> Implementing and empirically evaluating online recommendation algorithms for various models of adaptive agents is certainly an important direction for future research.
>
> ### Single-Agent Restrictions:
> Our model aims to capture specific phenomena relating to online recommendations, and is not intended to represent the full complexity of an applied recommender systems setting. We view our work as a step in the direction of representing more of this complexity via a bandit approach; our model is related to that in "dueling bandits" papers, which feature menus of recommendations for a single agent but do not incorporate adaptive preferences.
> In particular, we leave the aggregation of preferences across multiple users for future work. The single-agent setting is already quite rich when accounting for adaptive preferences. Deciding how to incorporate multiple agents in this setting is certainly an interesting question, and determining the appropriate model is not entirely straightforward.
>
> ### "Naive" Multi-Agent Extensions:
> While not practically efficient, one could imagine a Recommender using an separate instance of the algorithm in parallel for every Agent on the platform, and every Agent may have a distinct preference model and associated set of reward functions. Additionally, if Agent "types" correspond to their preference model but each Agent maintains a distinct history, we could use an instance of the algorithm for each type and speed up local learning by "batching" queries across Agents; after local learning is complete, the same regret bound could be obtained for the RC-FKM window, as we know each Agent's type and history. A more satisfying multi-agent model would likely involve a more elegant way of correlating preferences across users than entirely distinct preference models for discrete Agent types, which we are exploring for future work.
>
> ### Notation:
> The vector $s_{v}$ refers to the preferences scores of the Agent at the current memory vector $v$, as stated in Definition 1. We use $s_{v, i}$ to denote the score for the item at index $i$. $d_{TV}$ refers to the total variation distance (half the L1 norm), a common measure of distance between discrete distributions.
>
> ### Diversity vs. Entropy:
> The function of a "diversity constraint" is exclude to item distributions which place a large amount of mass on a small number of items. We note in Section 2 that, with the exception of Theorem 2, our positive results hold for any constraint set which is convex and contains the uniform distribution; these are natural requirements for any definition of a diversity constraint, as the uniform distribution is ``maximally diverse'' and taking a mixture of "sufficiently diverse" distributions should result in a sufficiently diverse distribution. We will make this clearer in the camera-ready version. We emphasize entropy because it is well-studied as a measure of dispersion for discrete distributions, as well as for its connection to the EIRD set that we show in Theorem 2.
>
> ### Other Comments:
> We address the limitations of our work in Section 5 by discussing remaining open questions, which we will expand for camera-ready, as well as through our negative results (Theorems 1 and 3) which show inherent difficulties of optimization in this setting.

---

### Meta-Review · Area_Chair_yrj7 · 2022-08-25

**Recommendation:** Accept
**Confidence:** Less certain

**Metareview:**

This paper studies a sequential recommendation problem where user preferences change over time based on the items selected by the user. The authors show that no sublinear regret algorithm exists for this setting. Under the constraint on preference diversity, they derive an algorithm with $O(T^{3 / 4})$ regret. The original ratings of the paper were 7, 7, 5, and 5; and they did not change after the rebuttal. The reviewers generally like the paper because it studies both adaptive users and diverse recommendations, while most other works try to avoid these topics. This is the strongest point of the paper and my recommendation for acceptance is based on that. On the other hand, the paper is overly theoretical and lacks experiments. Therefore, it is unlikely to appeal to a general recommender systems audience.

**Award:**

No

---

### Decision · Program_Chairs · 2022-09-14

Accept